

# Recent degradation of Interior Alaska permafrost mapped with ground surveys, geophysics, deep drilling, and repeat airborne LiDAR

Thomas A. Douglas[1,*], Christopher A. Hiemstra[1], John E. Anderson[2], Robyn A. Barbato[3], Kevin L. Bjella[1], Elias J. Deeb[3], Arthur B. Gelvin[1], Patricia E. Nelsen[1], Stephen D. Newman[3], Stephanie P. Saari[1], Anna M. Wagner[1]

[1]U.S. Army Cold Regions Research and Engineering Laboratory, 9th Avenue, Building 4070, Fort Wainwright, Alaska, USA 99709
[2]U.S. Army Geospatial Research Laboratory, Richmond, Virginia, USA
[3]U.S. Army Cold Regions Research and Engineering Laboratory, 72 Lyme Road, Hanover, NH, USA 03755

*Correspondence to: Thomas A. Douglas (Thomas.A.Douglas@usace.army.mil)

**Abstract.** Permafrost underlies one quarter of the northern hemisphere but is at increasing risk of thaw from
climate warming. Recent studies across the Arctic have identified areas of rapid permafrost degradation from both top-down and lateral thaw. Of particular concern is thawing of ice rich high carbon content syngenetic "yedoma" permafrost like much of the permafrost in the region around Fairbanks, Alaska. With a mean annual temperature of -2°C subtle differences in ecotype and permafrost ice and soil content control the near-surface permafrost thermal regime. Long-term measurements of the seasonally thawed "active layer" across central Alaska have
identified an increase in permafrost thaw degradation that is expected to continue, and even accelerate, in coming decades. A major knowledge gap is relating belowground measurements of seasonal thaw, permafrost characteristics, and talik development with aboveground ecotype properties and thermokarst expansion that can readily quantify vegetation cover and track surface elevation changes over time. This study was conducted from 2013-2020 along four 400 to 500m long transects near Fairbanks, Alaska. Repeat end of season active layer
depths, near-surface permafrost temperature measurements, electrical resistivity tomography (ERT), deep (>5m) boreholes, and repeat airborne LiDAR were used to measure top down thaw and map thermokarst development at the sites. Our study confirms previous work using ERT to map surface thawed zones, however, our deep boreholes confirm the boundaries between frozen and thawed zones that are needed to model top down, lateral, and bottom-up thaw. At disturbed sites seasonal thaw increased up to 25% between mid-August and early October
and suggests active layer depths must be made as late in the fall season as possible because the projected increase in the summer season of just a few weeks could lead to significant additional thaw. At our sites, tussock tundra and spruce forest are associated with the lowest mean annual near-surface permafrost temperatures while mixed



forest ecotypes are the warmest and exhibit the highest degree of recent temperature warming and thaw degradation. Thermokarst features and perennially thawed zones (taliks) have been identified at all sites. Our

measurements, when combined with longer-term records from yedoma across the 500,000 km$^2$ area of central Alaska show widespread initiation of near-surface permafrost thaw since roughly 2010. Using this partial area of the yedoma domain and projecting our thaw depth increases, by ecotype, across this domain we calculate 0.44 Gt of permafrost soil C have been thawed over the 7 year period, an amount equal to the yearly $CO_2$ emissions of Australia. Since the yedoma permafrost and the variety of ecotypes at our sites represent much of the Arctic and

subarctic land cover this study shows remote sensing measurements, top-down and bottom-up thermal modelling, and ground based surveys can be used predictively to identify areas of highest risk for permafrost thaw from projected future climate warming.



# 1 Introduction

Permafrost underlies ~40% of central Alaska, a 500,000 km$^2$ region stretching east to west from the Canadian

border to the Seward Peninsula and north to south from the Brooks Range to the Alaska Range. This is expected to mostly disappear from the near surface (upper 1 m) by 2100 (Pastick et al., 2015). Mean annual temperatures in interior Alaska, currently roughly -2°C (Jorgenson et al., 2020), are projected to increase by 2°C by 2050 (Douglas et al., 2014) and 5 °C by 2100 (Lader et al., 2017). Roughly half of the discontinuous permafrost in the area represents late Pleistocene ice and organic carbon rich "yedoma" (Kanevskiy et al., 2011; Strauss et al.,

2016). In total, yedoma contains almost a third of the permafrost carbon pool despite underlying only 625,000 km$^2$ of central Alaska and Russia, ~7% of the total global permafrost land area (Heslop et al., 2019). Yedoma permafrost contains large organic carbon stocks that are extremely biolabile (Vonk et al., 2013; Strauss et al., 2017; Heslop et al., 2019) and highly vulnerable to thaw due to high ice content and the prevalence of massive ice bodies (Strauss et al., 2013; 2017).

Throughout much of central Alaska the permafrost is undergoing widespread top down and lateral thaw (Jorgensen et al., 2013; 2020; Douglas et al., 2020; Circumpolar Active Layer Monitoring Network, 2020). Recent measurements of lateral thaw (Neumann et al., 2019) and modelled bottom up thaw (McClymont et al., 2013; Way et al., 2018) of discontinuous permafrost bodies have also been reported. Permafrost degradation alters hydrogeology, soils, vegetation, and microbial communities (Racine and Walters, 1994; Walker et al., 2006;

Mackelprang et al., 2011; 2017; Wilhelm et al., 2011; Wolken et al., 2011; Messan et al., 2020). Microbiological and trace metal processes are also likely to change in thawing permafrost ecosystems due to alterations in soil, vegetation, and wetland properties (Grosse et al., 2011; Douglas et al., 2013; Schuster et al., 2018; Burkert et al., 2019). In addition to these ecological and hydrologic changes, permafrost degradation presents an expensive and uncertain challenge for the design, siting, and maintenance of vertical and horizontal infrastructure in cold regions

(Hjort et al. 2018).

The thermal state of near surface permafrost is controlled by topography, slope, aspect, soil texture, ground ice content, air temperature, hydrology, land cover, snow depth and timing, and liquid precipitation (Osterkamp and Romanovsky, 1999; Jorgenson and Osterkamp, 2005; Myers-Smith et al., 2008; Loranty et al., 2018). In relatively warm areas like Interior Alaska, the permafrost is "ecosystem protected" (Shur and Jorgenson, 2007) by an

insulating organic-rich soil, plant litter, and vegetation surface layer. Disturbance to this insulating layer from



climate warming, infrastructure development, or wildfire increases ground heat flux and promotes top down, lateral, and bottom-up thaw (Viereck et al., 1973; Yoshikawa et al., 2003; Nossov et al., 2013).

Commonly, the first signal of an altered permafrost thermal state is an increased seasonally-thawed "active" layer (Hinkel et al., 2003; Shiklomonov et al., 2010). Seasonal trends in active layer depth, particularly across a variety

of ecotypes, can provide information on how and where permafrost degradation features initiate and expand. Low-ice content dry sandy soils typically have deeper active layers than ice rich silt or organic-rich soils (Brown et al., 2015; Loranty et al., 2018). As such, active layer measurements can infer information about subsurface soil characteristics. When top-down permafrost degradation occurs, the active layer depth may increase before any readily identifiable change in surface vegetation or geomorphology occurs. The most pronounced terrain surface

features form when thaw of ice rich permafrost leads to thermokarst (hollows formed by ground subsidence following thaw of ice-rich permafrost; Kokelj and Jorgenson, 2013; Brown et al., 2015; Douglas et al., 2016). Thermokarst features include lakes, bogs, fens, and pits in lowlands and thaw slumps and active-layer detachments in uplands (Smith et al., 2005; Jorgenson et al., 2013).

There is a need to broadly apply remotely-sensed analyses to identify high ice content permafrost at risk of top

down and lateral thaw degradation to support ecological, hydrologic, and engineering, investigations. Identifying risk factors for thermokarst initiation typically requires combining ground-based surveys and remotely-sensed measurements. Where permafrost is associated with surface biophysical characteristics that can be measured remotely, standoff detection tools like airborne LiDAR and repeat imagery analysis can be applied toward tracking trajectories of change over large regions (Jones et al., 2013; Chasmer and Hopkinson, 2016; Lewkowicz and Way,

2019). Geophysical techniques, predominantly electrical resistivity tomography (ERT), have been recently coupled with airborne and active layer measurements to detect thermokarst development and associate ice content with terrain geomorphology (Yoshikawa et al., 2006; Douglas et al., 2008; Lewkowicz et al., 2011; Hubbard et al., 2013; Minsley et al., 2015; Bjella, 2020) and biophysical characteristics (Douglas et al., 2016) at broader scales. A combination of repeat active layer measurements, geophysical surveys, and airborne LiDAR have been

used to map subsurface permafrost bodies, quantify top-down thaw, and identify locations where thermokarst features have been initiated or expanded (Douglas et al., 2016; Rey et al., 2020). Long-term ground-based time series measurements can be combined with ERT to quantify top down thaw, track the initiation and lateral expansion of thermokarst features, and identify where ecosystem characteristics influence the permafrost thermal regime. Further, extents of the base and sides of discontinuous permafrost bodies with geophysical measurements

confirmed with deep boreholes is needed to monitor and better model lateral and bottom-up thaw.





The objective of this study was to establish relationships between ecotype, permafrost soil characteristics, and seasonal thaw across a variety of terrains in interior Alaska. Our sites represent 159,000 km² of high ice content yedoma permafrost with massive ice wedges that is present across the 500,000 km² expanse of central Alaska. We made repeat seasonal and active layer thaw depth measurements, performed electrical resistivity tomography, and characterized the permafrost with boreholes up to 15 m deep along transects that represent the five most common ecotypes associated with central Alaska's yedoma permafrost. Ground-based information was combined with high-resolution repeat airborne LiDAR imagery to identify thermokarst initiation and quantify terrain elevation changes over time. Long term active layer depth measurements across central Alaska were used to place our measurements into spatially broader and temporally longer scales. We used our active layer-ecotype relationships to model the amount of yedoma permafrost carbon that has thawed across central Alaska since 2013. The goals of this work were to measure seasonal thaw and thermokarst development over time and identify surface and subsurface terrain properties that can be remotely quantified, like vegetation type and thaw subsidence, with permafrost geophysical, soil, and active layer characteristics.

## 2. Field Measurements

### 2.1 Study location and Site Descriptions

Our field sites are located near Fairbanks, Alaska (Fig. 1). The region has a continental climate with a mean annual air temperature of -2.4ºC, typical mean summer temperatures of 20°C, mean winter temperatures of -20°C, and yearly extremes ranging from 38°C to -51°C (Jorgenson et al., 2001; 2020). Mean annual precipitation is 28.0 cm (Wendler and Shulski, 2009) with a typical annual snowfall of 1.7 m (Jorgenson et al., 2001) that represents 40-45% of the annual precipitation (Liston and Hiemstra, 2011). Discontinuous permafrost features in the area are up to 60 meters thick and are located primarily in lowlands, along north-facing slopes, and where soils or vegetation provide adequate thermal protection (Racine and Walters, 1994; Jorgenson et al., 2008; Douglas et al., 2014). Permafrost at our field sites is Pleistocene syngenetic ice-rich "yedoma" formed through repeated deposition of windblown loess and organic matter (Shur and Jorgenson, 2007; Douglas et al., 2011; Strauss et al., 2016). Almost a third (181,000 km²) of the global yedoma permafrost is in Alaska and of that the majority is in a swath of central Alaska between the Brooks Range to the north and the Alaska Range to the south (Strauss et al., 2016). Carbon content in the permafrost of 2–5% (~10 kg/m³) is up to 30 times greater than unfrozen mineral soil (Strauss et al., 2013).



Our field investigations were organized along four transects crossing a variety of lowland (three) and upland (one)
permafrost landscapes (Fig. 1). The 400m "Farmer's Loop 1" and 500 m "Farmer's Loop 2" transects were located
at 64.877 °N, 147.674 °W and 64.874 °N, 147.677 °W, respectively. These two transects cross a variety of
ecotypes including mixed deciduous forest (dominated by *Picea glauca* and *Betula neoalaskana*), *Salix spp.*
riparian wetland, *Eriophorum vaginatum* tussock tundra, and moss- black spruce (*Picea mariana*) forest. Trail
crossings and other clearings (disturbed areas), devoid of trees, punctuate the transects in multiple locations and
are inhabited by the grass *Calamagrostis canadensis*. A nearby Circumpolar Active Layer Monitoring (CALM)
site has a 16-year active layer record (CALM, 2020). A 500 m transect at the nearby Creamer's Field Migratory
Waterfowl Refuge (64.868 °N, 147.738 °W) transitions from mixed deciduous forest (*Betula neoalaskana* and *P.
glauca*) in the first 150 m, before entering moss- black spruce (*P. mariana*) forest for ~50 m. Farther north on this
transect *Eriophorum vaginatum* tussock tundra is prevalent with isolated *B. neoalaskana* and *P. mariana* trees
along with two east-west oriented trail crossings. A 400 m southwest-northeast oriented transect was also
established above the CRREL Permafrost Tunnel in Fox, Alaska (64.950 °N, 147.621 °W). Vegetation at this site
transitions from black spruce (*P. mariana*) forest with S*phagnum* moss through 1960s-era clearings and trails and
shrub-dominated (*Rhododendron groenlandicum* and *Betula nana)* clearings into Glenn Creek's riparian zone.
Our field sites encompass common boreal ecoregion land cover classes (sub-polar needleleaf and deciduous forest,
mixed forest, shrubland, grassland; wetland, barren; disturbed; and water). Together, these classes account for
74% of the boreal ecoregion's area in North America (Latifovic et al., 2017).

## 2.2 Satellite and LiDAR imagery

Imagery was needed to examine transect land cover types corresponding with LiDAR data. Cloud-free high-
resolution Maxar WorldView-2 (WV 2) satellite imagery (2 m multispectral; 0.5 m panchromatic) was obtained
for all sites (Fig. 1) on 7 June 2020. The images were orthorectified and pan-sharpened using ENVI's 5.5.3 SPEAR
pan-sharpening (Gram-Schmidt) to qualitatively maximize the imagery to match LiDAR data. Airborne LiDAR
measurements are helpful for monitoring surface changes and examining surface roughness characteristics.
LiDAR data collected in 2010 were only available for the Creamers Field Transect (Hubbard et al., 2011). These
data were collected by aircraft 5-6 May 2010 using an Optech ALTM Gemini (Toronto, Canada) 1064 nm LiDAR
with a pulse frequency of 70 hHz and 12° scan angle. The resulting point cloud (point density = 4 points/m$^2$) was
used to create a 1.21 m resolution DSM with a vertical accuracy of 16 cm. All three of the sites (Fig. 1) were
scanned with a different LiDAR platform 17-18 May 2020. An aircraft-mounted Leica (Wetzlar, Germany)
ALS80 (1064 nm) acquired surface returns at an average density of ≥25 points/m$^2$. In both the 2010 and 2020



collections, aircraft and sensor position and attitude data were indexed by GPS time for post-processing correction

and calibration. Measurement accuracy yielded a root mean square error (RMS) of ≤6.6 vertical cm in 2020. Point clouds were processed to create hydro-flattened raster surfaces with a spatial resolution of 0.25 m. Hydro-flattening was used to remove errant point cloud elevation artefacts from resulting DEMs given water's low reflectance. In this process, stream, pond, and lake boundaries were identified and the DEM was corrected to more accurately portray water level elevation given the identified shoreline, yielding a waterbody-smoothed product.

Changes in elevation at the Creamer's Field site between 2010 and 2020 were calculated by subtracting 2010 elevations from 2020 elevations (2020 minus 2010) using raster algebra to delineate elevation losses (negative values) and gains (positive values) over the 10 year period.

**2.3 Field survey measurements, coring, and meteorology**

In May-June 2013, 1-m wide trails were delicately hand cleared of large woody vegetation along linear transects to improve access for repeat surveying and geophysical measurements. A Trimble (Sunnyvale, California USA) R8 DGPS was used to survey pin flag measurement markers at a 4 m spacing along each of the four transects. We used a 1 cm diameter 1.7 m long graduated metal rod ("frost probe") to make seasonal thaw depth measurements at each flag location to quantify the end of summer season "active" layer (Shiklomonov et al., 2013).

Measurements were repeated in mid-October from 2013 to 2020, however, in 2014 additional measurements were made in June, July, and August . Active layer measurements from 2013 to 2017 were published previously (Douglas et al., 2020).

A Geoprobe 7822 Direct Push Technology track mounted drill rig was used to collect deeper (between 4 and 15.6 m deep) cores in late winter and spring 2014. Coring was limited to locations that had trail access for the heavy

tracked vehicle. Three wood fragments found in cores from the Farmer's Loop transects were analysed for $^{14}$C age and $\delta^{13}$C measurements following commonly used radiocarbon analytical procedures at Beta Analytic (Miami, Florida). A SIPRE corer was used to collect 2 to 3 m deep cores in the spring of 2017 and 2018 at locations representing the major ecotypes at each site following established methods (Douglas et al. 2011). Gravimetric (Geoprobe) or volumetric (SIPRE) moisture contents were measured following established methods (Phillips et

al., 2015).

An Onset (Bourne, Massachusetts, USA) HOBO U23 Pro v2 external temperature and relative humidity logger with a solar radiation shield was installed 2 m above the ground surface at the Permafrost Tunnel and Farmer's

Loop sites. Onset HOBO U23 Pro v2 two channel external temperature loggers were installed at depths of 1.2 m at nine locations across our field sites. The thermistor was protected by a plastic sleeve and installed into the ground after a 0.75 cm diameter hole was excavated using a slide hammer and rod.

## 2.4 Electrical resistivity tomography

We used a "SuperSting" R8 eight channel portable induced polarization galvanic earth resistivity meter (Advanced Geosciences Incorporated, Austin, Texas) for ERT measurements. ERT measurements were conducted in mid-summer 2013 using six cables, each with 14 take-out electrodes. Our electrode spacings of 2.5 to 4 m. achieved a maximum subsurface penetration depth of ~30 m. We used a dipole-dipole array because it represents spatial aspects of ice-rich terrain and provides horizontal resolution sufficient for detecting vertical structures in permafrost (Kneisel, 2006; Douglas et al., 2016). Contact resistance was measured at each electrode prior to initiating the survey to ensure cable connectivity. At rare instances when contact resistances were higher than 2,000 Ω-m salt water was added around the electrode and contact resistance was re-measured until resistance fell below 2,000 Ω-m. Electrodes were typically 45 cm long but electrodes up to 3 m in length were used in areas with thick vegetation mats or moss.

We used RES2DINV (Geotomo Software, Penang, Malaysia) to perform two-dimensional model interpretation. The software provides signal smoothing and constrains inversion with finite difference forward modeling and quasi-Newton techniques (Loke and Barker, 1996; Loke et al., 2003). A least-squares inversion achieves convergence by comparing changes in root mean squared (RMS) quadratic error between two and five iterations, then three and five iterations, etc. Convergence was achieved when RMS error values reached ~10% convergence and further iterations would not significantly lower the RMS values.

## 3. Results

### 3.1 Satellite and LiDAR imagery

The four transects we studied contain five dominant ecotypes of the boreal biome and central Alaska yedoma terrain (mixed deciduous forest, wetlands, tussock tundra, moss- black spruce forest with a thick moss cover, and disturbed; Douglas et al., 2020). In total, these classes account for 74% of the boreal ecoregion's area in North America (Latifovic et al., 2017). The Creamer's field transect starts in birch forest, transitions to mixed forest, and at 140 m it transitions abruptly to tussock tundra for the remainder of its 500 m length (Figs. 2a and 3a).





Patterned ground (near-surface ice wedge polygons) is readily evident in the airborne LiDAR in the mixed forest along the first ~150 m of the transect (Fig. 2b). This area is characterized by baydzherakhs ("Siberian cemetery mounds" that remain when ice wedges melt) up to 2 m higher than the adjacent degraded ice wedges. When the transect transitions to tussock tundra, ice wedge polygons are no longer as strongly visible at the ground surface, yet, polygonal ground is evident in true color satellite imagery and airborne LiDAR throughout the remainder of

the transect (Figs. 2a and 3a). Winter trails, dominated by native graminoids, are evident in the true color and LiDAR images at 290 m and 460 m.

There is subtle evidence of ice wedge polygons along the two transects at the Farmer's Loop field site (based on WV 2 satellite images and LiDAR; Figs. 4a, 4b, 5a, and 5b). Both transects start in mixed forest that extends for ~120 m. Transect 1 crosses a small wetland feature at 80 m before transitioning to tussock tundra until 310m.

After a graminoid-dominated trail the ecotype changes to *Picea mariana* (black spruce) forest. Farmer's Loop transect 2 shifts from mixed forest (*Betula neoalaskana*, *Salix* spp., and *Picea glauca*) to a flow through fen wetland from 120 to 170 m. After a trail crossing at 200 m the ecotype shifts abruptly to tussock tundra until the 400 m mark where a trail crossing separates the tussock tundra from mature *Picea mariana* (black spruce) forest.

Though ice wedges are present throughout the 300 m of subsurface passages that run partially below the

Permafrost Tunnel transect they are covered by a ~5 m thick surface layer of Holocene silt (Hamilton et al., 1988) and a thick veneer of mosses, lichen, shrubs, and trees. Polygonal ground is not identifiable in visible (WV 2) or LiDAR imagery at the surface along the Permafrost Tunnel transect (Figs. 6a and 6b). The Permafrost Tunnel Transect originates in spruce forest, transits through shrubland, and crosses trails and Glenn Creek before entering black spruce forest again. Numerous trail crossings identified as disturbed locations and a large thermokarst

feature near Glenn Creek are also present. Anthropogenic features (i.e. disturbances) like roads, trails, and clearings, are easily identifiable in the satellite imagery at all sites.

Our longest LiDAR data series spans May 2010 to 2020 and is limited to the Creamers Field transect (Fig. 7). A difference in those elevations (2010 elevations were subtracted from 2020 elevations) indicates substantial elevation losses (over 1 m of subsidence) along the northern edge of degrading polygon ice wedges in an area that

has transformed into an elongate lake. This thaw front of degrading permafrost is evident along the southern margin of the transect (left side of the map). Water levels in the pond and in some ice wedge polygon troughs show higher elevations in 2020 compared to 2010 (0.2 to 1.0 m) due to deeper and more persistent precipitation in the last three years..



### 3.2 Field survey measurements, coring, and meteorology

The mid-June and early August seasonal thaw depth measurements and October active-layer measurements in 2014 show a steady downward movement of the thaw front throughout the summer season (Figs. 2c, 3c, 4c, 5c, and 6c). The majority (~80%) of the summer season thaw at the tussock tundra and spruce forest sites occurs by early August. However, in the wetland, disturbed, and mixed forest ecotypes the increase in thaw depth from early August to mid-October is up to one third of the eventual active layer depth. This is particularly evident along the

first 200 m of the Farmer's Loop 2 transect (Fig. 5c). The wetland, disturbed, and mixed forest ecotypes yield the deepest active layers at all sites. The tussock tundra and spruce with moss ecotypes consistently yield the lowest seasonal thaw measurements and show little change between August and October. A statistical summary of the active layer depths is provided in Table 1. It is clear that for all five ecotypes active layer depths increased substantially between 2013 and 2020. Since each ecotype is associated with different starting thaw depths the

percent increase in depth between 2013 and 2020 is a good indicator of top down thaw over the seven year period. The tussock, wetland, disturbed, and mixed forest ecotypes all exhibited increases in active layer depth of more than fifty percent while the increase in the spruce forest was 33 percent. Across all five ecotypes the mean of active layer depth in 2013 was statistically significantly smaller than the measurement in 2020.

We collected 14 deep (greater than 5 m) cores with the Geoprobe and 12 shallow (3 m or less) SIPRE cores (Figs.
2d, 3d, 4d, 5d, and 6d). Frozen bulk density (SIPRE cores; frozen mass divided by volume) and gravimetric moisture content (Geoprobe cores; mass of water lost through drying divided by volume) from the cores are included in Table S1. In general, the upper most core samples, which consisted of surface vegetation and organic matter, yielded the greatest moisture content values. Most of the cores had peat or organic and ice rich permafrost in the upper 1-3 meters and along some deeper sections and these typically yielded gravimetric moisture contents

greater than 100%. The deeper permafrost soils were characterized as ice rich and ice poor silts and sands with gravimetric moisture contents of 60-150% which is similar to measurements of syngenetic yedoma type permafrost in the Permafrost Tunnel (Bray et al., 2006; Douglas et al., 2011; Douglas and Mellon, 2019). For the subset of SIPRE cores that yielded frozen bulk density values they ranged predominantly between 900 and 1400 kg/m³ which is also similar to values from the nearby Permafrost Tunnel, however, some ice rich and peaty zones

yielded values above 1400 kg/m³.

Coring allowed us to confirm that large decreases in apparent resistivity values from ERT confirmed 0°C isotherm boundaries between frozen and thawed material. Notably, a deep core on the Farmer's Loop 1 transect (358 m)





collected thawed silt from 9.15-10.35 m. The ERT measurements at that location show a large thawed region starting at ~10 m depth identified by resistivity values of ~1,000 Ω-m (more discussion below). Along the sides

of most of the thawed zones we identified at each site marked changes in apparent resistivity allowed identification of the lateral boundaries of thermokarst features. Notably, these include thawed zones in the baydzherakhs along the beginning of the Creamer's Field transect, thermokarst pits along the Farmer's Loop and Permafrost Tunnel transects, a large (~50m) lateral expansion of the thermokarst toward the end of the Permafrost Tunnel transect, and thawed regions below numerous disturbed areas at all sites. Some of the 2017

and 2018 SIPRE cores identified seasonal taliks between the bottom of the winter frozen layer and the top of the permafrost (Table S1). For example, a 2017 core from the mixed forest at 87 m along Farmer's Loop transect 1 yielded thawed silt from to 46 to 102 cm. Cores collected in 2018 along Farmer's Loop transect 2 in mixed forest at 71 m collected thawed silt from 84-117 cm and two cores from the Permafrost Tunnel (at 52 and 305 m) identified thawed silt from 31-49 cm and 45-70 cm, respectively. These thawed zones are located above the typical

active layer depth for those locations and indicate the winter freeze did not extend downward to the top of the permafrost.

We obtained $^{14}$C ages from wood fragments collected from three Geoprobe core samples. An age of 10,360 +/- 360 y ($\delta^{13}$C: -27.7 ‰) was measured at a depth of 1.02 m in the tussock area at 306 m on Farmer's Loop transect 1. At 358 m along the same transect and also in the tussock area at 0.67 m depth the $^{14}$C age was 10,160 +/-160 y

($\delta^{13}$C: -28.0 ‰). Along the Farmer's Loop 2 transect, in the spruce forest at 420 m and at a depth of 0.49 m depth a wood fragment yielded a $^{14}$C age of 7,200 +/-190 ($\delta^{13}$C: -28.7 ‰).

Air temperature, wet (rain) and dry (snow) precipitation, and the snowpack depth, from April 1, 2013 through October 31, 2014, are provided in Fig. 8. This time period encompasses when the repeat summertime thaw depths and geophysical analyses were measured. The meteorological measurements were made by the National Weather

Service at the Fairbanks International Airport (PAFA) 8 km to the southwest of the Creamer's Field and Farmer's Loop transects and 17 km southwest of the Permafrost Tunnel transect. Air temperatures at the Permafrost Tunnel and Farmer's Loop sites are also included and they did not deviate substantially from one another or from the 90 year PAFA mean during the timeframe of the study. Air temperatures went above 0°C around the middle of May and did not go below freezing again until late October. The summer of 2013, when our ERT measurements were

made, experienced a total of 14.5 cm of wet precipitation which is slightly lower than the 90 year mean of 18.5 cm. The 2013 summer mean temperature of 12.2 °C was close to the 90 year summer mean of 11.8 °C. (Jorgenson et al., 2020). In terms of heating degree days the summer of 2013 (1133) was slightly above the historical mean



of 1090. The winter of 2013-2014 total snowfall of 1.22 m was slightly below the historical mean of 1.7 m. The wet precipitation total for the summer of 2014 (37.1 cm) was anomalously higher than the mean.

Fig. 9 provides mean annual ground temperatures (MAGT) at 1.2 m depth for our field sites representing the range of ecotypes at our sites. When they were installed in 2013 this depth represented the upper 40-60 cm of permafrost at our sites. We also provide the mean annual ground temperature at 1.2 m depth for each site between October 1, 2013 and October 1, 2019 (Table 2). The Creamer's Field disturbed site, in a clearing devoid of permafrost 114 m southeast of the Creamer's Field transect, is the only location where the mean annual soil temperature at 1.2 m

is above freezing (4.31 °C). The remaining thermistors emplaced to 1.2 m depth are located in permafrost and their temperatures remained below 0 °C for the entire record, however, they all show warming between 2013 and 2019. Thermistors in the mixed forest and spruce forest ecotypes have been steadily approaching 0 °C. Mean annual permafrost temperatures at the two mixed forest sites (-0.24 and -0.20 °C) and spruce forest (-0.25 °C) are substantially higher than for the three tussock tundra thermistor locations (-1.55, -1.39, and -2.60 °C). The tussock

tundra sites also have the lowest winter permafrost temperatures.

### 3.3 Electrical resistivity tomography

Electrical resistivity tomography measurements across the transects (Figs. 2e, 3e, 4e, 5e, and 6e) provide insight into the presence or absence of permafrost at depths of up to 40 meters below the ground surface, particularly when they are calibrated with subsurface information from boreholes. Resistivity values of ~800 Ω-m and higher

have been reported for syngenetic permafrost near Fairbanks while values below 800 Ω-m are generally assumed to represent thawed material (Hoekstra and McNeill, 1973; Harada et al., 2000; Yoshikawa et al., 2006; Douglas et al., 2008; 2016; Minsley et al., 2016). For the Creamer's Field transect (Figs. 2 and 3) the first 130 m consists of actively degrading ice wedge polygons that are present along a slightly-elevated bench (~3 m) in mixed forest. In this area, pockets of low resistivity material in the upper 1-2 m denote the thawed areas around the

baydzherakhs. Permafrost (1.2 m depth) temperatures at this location have increased steadily since 2013 and are currently are only slightly below freezing (Fig. 9). This is the same region that exhibits the largest rates of ground subsidence in repeat LiDAR differencing at the site (Fig. 7). The greatest active layer depths along the Creamer's Field transect are located in this area. It is noteworthy that at some locations in this area of the transect the active layer expanded ~25% between late August and early October. Clearly the permafrost along the first ~75 m of the

transect is undergoing active thaw degradation. At 140 m along the transect the ground surface drops slightly and from there to the end of the transect the ecotype is characterized as tussock tundra. The permafrost in the tussock





region is colder (MAGT at 1.2 m: -1.55°C) with little downward expansion of the active layer between late August and early October. The lone exception is a small (2-5 m across) thermokarst feature at a transect distance of 380 m that exhibits anomalous thaw throughout the summer and is barely discernible in the aerial image, LiDAR, and
ERT measurements.

Both of the Farmer's Loop transects (Figs. 4 and 5) start in a mixed forest for the first ~125 m. In this area the near surface permafrost is comprised of silts with minor peat and some layers with low gravimetric moisture contents of ~23-50 g/g. These areas also have the deepest active layer depths. Some small thawed areas as well as disturbed areas and small wetland features are identifiable by resistivity values of 200-700 Ω-m in the upper few
meters and active layers more than a meter deep. As with the Creamer's Field site these ecotypes are associated with the largest increase in active layer depth between late August and early October. Permafrost in this area appears to be present for only the upper 15 to 20 m. A small flow through fen from 120 to 180 m along Farmer's Loop transect 2 has a low resistivity region ~15 m below the ground surface. These areas are associated with the warmest permafrost temperatures at the Farmer's Loop site (-0.05 °C at 1.2 m; Fig. 9; Table 2).

Once the transect transitions into tussock tundra the active layer depths become shallower and the temperatures at 1.2 m decrease. Regions of anomalously low ERT measurements are associated with small surface thermokarst features, water features, or disturbed areas. Soils in the area are characterized as silts with varied ice contents. A deep drill hole at 358 m along Farmer's Loop transect 1 identified thawed material starting at 915 cm and this corresponds with the marked decrease in resistivity values at this same location and depth.

At the permafrost tunnel site (Fig. 6) the ERT resistivity values are 1,000 to 2,000 Ω-m in the upper ~4 m with a repeating pattern of markedly higher values (5,000 to 10,000 Ω-m) from 4 to 10 m in depth and at a ~10 m spacing. This is consistent with ice wedge polygon structures in the subsurface, likely the "Upper Silt Unit" overlain by Holocene silts as mapped by Hamilton et al. (1988) and corroborated by our core drilling at the site. These subsurface ice wedge structures do not extend to the surface and, as such, they do not relate to vegetation type, or
seasonal thaw depths changes across the transect. Towards the far end of the Permafrost Tunnel transect the ERT from 2014 identifies the lateral edges of a large thermokarst feature forming at the site. Thaw between 2014 and 2020 has added roughly 25 m of width to both sides of the feature.



## 4. Discussion

The results from this study clearly show, through a variety of corroborating measurements, ice rich yedoma
permafrost in the area around Fairbanks, Alaska has been warming and actively degrading in numerous locations.
Four major lines of evidence show permafrost thaw degradation has been initiated and is likely increasing at our
sites. First, active layer measurements show thaw depths have been increasing across all ecotypes since 2013
(Table 1), however, some ecotypes experience deeper seasonal thaw than others. In 2014, the only year when we
made repeat thaw depth measurements during the summer, (Figs. 2-6 c and d) it is apparent that in mid-June there
is minimal variability in thaw depth except for some of the disturbed areas that eventually exhibit the deepest
active layer thickness. By mid-July the disturbed and mixed forest ecotypes exhibit the most seasonal thaw and
these ecotypes have the deepest end of season active layers. In some locations, particularly the disturbed sites,
mid-August seasonal thaw depths and early October active layer measurements show a ~20% increase over that
6 week period. The length of the summer growing season in the area has increased by 38 days (Wendler and
Shulski, 2009) and our thermistor measurements (Figure 9) show peak soil temperatures at 1.2m typically occur
in late November. From this, it is clear that the timing of thaw depth measurements may have to be pushed later
into the fall to adequately represent the summer season, particularly in mixed forest and wetland ecotypes (Figures
2-6). This has ramifications for field studies where thaw depth measurements are made in August during the end
of most field seasons. Increasingly, these measurements will not represent the true active layer depth.

Previous studies have established vegetation provides a range of ecosystem protection properties for permafrost
(Shur and Jorgenson, 2007; Loranty et al., 2018). Recent measurements confirm this and identify strong links
between different ecotypes and top-down thaw of permafrost in Interior Alaska (Yi et al., 2018; Douglas et al.,
2020; Jorgenson et al., 2020; Kropp et al., 2020). In this study, some ecotypes are associated with consistently
deeper active layer measurements over time. Disturbances, like trail crossings, are associated with dramatically
deeper seasonal thaw than any other ecotypes and some of them are also expanding laterally. Removal or alteration
of the organic soil layer or moss ground cover increases the ground heat flux and promotes more rapid seasonal
and permafrost thaw (Nicholas and Hinkel, 1996) due to the loss of the "ecosystem protection" of permafrost in
the area (Shur and Jorgenson, 2007). In many locations, active layer depths have increased since 2013 to greater
than 2m which is greater than typical winter freezeback. Infrastructure development and wildfire are the most
likely ways for land cover to change to a disturbed ecotype. Post-fire forest succession to a mixed forest, which
is increasingly occurring across Interior Alaska and much of the boreal biome, will also undoubtedly lead to
warmer surface soils and more top-down permafrost thaw (Kasischke and Johnstone, 2005; Johnstone et al., 2010;



Jafarov et al., 2013; Brown et al., 2015). Tussock tundra and some of the spruce forest sites yield the shallowest active layer depths. As such, if vegetation were to change from tussocks or spruce to a mixed forest or disturbed

(i.e. no moss or forest vegetation) land cover the potential risk of top-down permafrost thaw increases considerably.

Our results support recent work at our study sites that show the disturbed, mixed forest, and wetland ecotypes exhibit the deepest active layers (Douglas et al, 2020). That study presents measurements from 2014 to 2017 at the same sites presented here and links deeper active layer depths with wetter summers. The four additional years

presented here show the thaw front has continued to migrate downward despite the lack of anomalously wet summers in 2018 and 2019. At most sites the 2020 active layer depths are the deepest in the record and the comparatively shallower thaw depths measured in 2013 have not been repeated at any site since then.

The increase in active layer depths we measured at our sites since 2013 is similar to the longer-term trend represented at all six Circumpolar Active Layer Monitoring sites spread across 500,000 km$^2$ of central Alaska (the

east-west swath south of the Brooks Range and North of the Alaska Range; Figure 10; CALM, 2020). At most sites a steady increase in active layer depth was initiated around 2010 and has continued since. Within this region of central Alaska 159,000 km$^2$ of the permafrost has been identified as yedoma. Using the regions mapped as yedoma as a "cookie cut" from ecotype maps that cover all of central Alaska (Jorgenson and Meidlinger, 2015; Raynolds et al., 2019) we calculate the five ecotypes in our study represent 90% of the total land cover on top of

yedoma permafrost. Mixed forest (34%), moss-spruce forest (23%), wetland (20%), and tussock tundra (13%) are widely prevalent while disturbed sites are not identified, likely due to their small spatial scale. Using the 2013-2020 increase in active layer from our field measurements, projecting the area of each ecotype across the central Alaska domain mapped as yedoma, and using the mean organic carbon concentration of 10 kg/m$^3$ from Strauss et al. (2013) we calculate the total central Alaska yedoma permafrost organic carbon (OC) pool that has thawed since

2013 as 0.44 Gt OC. This does not include non-yedoma permafrost in the same region that likely has also thawed since 2013.

Additional evidence indicating thaw of near-surface permafrost at our sites includes some thermistor measurements showing at 1.2 m depth approaching and eventually warming above 0°C (Fig. 8) at some sites. Mixed forest sites have warmed the most and all three of our 1.2 m deep thermistors in this ecotype exhibit a

steady warming that has been retarded at ~-0.1°C, likely due to latent heat effects in the transient layer and below (Boike et al., 1998; Shur et al., 2005). The tussock and spruce forest ecotypes do not show the steady increases in



permafrost temperatures, however, the overall trend in mean annual temperatures at 1.2 m depth at these sites is increasing (Table 2).

The third indication of near-surface permafrost degradation is the widespread development of supra-permafrost
taliks (an unfrozen layer between the active layer and the top of the near surface permafrost) zones developing) at our sites indicated by SIPRE cores collected in 2017 and 2018. At many locations, the seasonal thaw has proven to be deeper than the depth of winter freeze-back. The seasonal taliks we found are located predominantly in the mixed forest and disturbed ecotypes. These areas contain the warmest near surface permafrost and in some cases the low ice content sandy silts have a higher thermal conductivity that promotes the movement of heat into the
ground. At some of these sites we have had to augment thermistors at 1.2 m depth by installing deeper (i.e. 2 to 2.5 m thermistors) to maintain measurements of the near surface permafrost temperatures when the thaw front reaches 1.2 m. Since thaw depths increased in 2019 and 2020 it is likely these taliks have expanded vertically and laterally.

Numerous lines of visual evidence provide the fourth indication of active permafrost thaw in our research area.
The most dramatic is that of the ground subsidence (thermokarst) associated with permafrost thaw in the mixed forest region of the Creamer's Field transect (Fig. 9). Ice wedge polygons in the area have warmed steadily since 2013 and repeat LiDAR analysis shows baydzherakh development has expanded due to melting ice wedges (Fig. 7). This suggests potential hydrologic and soil thermal process changes are ongoing in that area (Liljedahl et al., 2016). Anomalous thaw depths coinciding with recent development of thermokarst pits are evident in airborne
LiDAR and true color images at all of our sites. At the Creamer's Field site degradation of ice wedge polygons is evident in our repeat LiDAR analysis. The most dramatic thaw subsidence occurred in the mixed forest area represented by the first 150 m of our transect. In this area near surface permafrost soils, comprised of lower ice content silts and sands, are warmer and baydzherakhs were already forming when we initiated our study. However, the low lying troughs between polygons have dropped by 1-1.5 m in the decade from 2010 to 2020. Further out
this transect in some select areas it appears ground elevations of the polygon troughs increased between 2010 and 2020. We attribute this to this area being extremely low lying and more standing water being present in the troughs from snowmelt in May, 2020 compared to 2010. Due to the standing water we cannot ascertain whether the ice wedges in this area have melted at all.

Repeat yearly active layer depth measurements show the thermokarst features have been extending vertically
downward and horizontally since 2013. Examples are at the Creamer's Field transect at ~100m; Farmer's Loop

transect 1 at 68 m and 360 m; Farmer's Loop transect 2 at 8 m, 88 m, 116 m, and 408 m; and the Permafrost Tunnel transect at 64 m, 10 8m, 140 m, and particularly at 328 m. These thermokarst features are forming either in areas of mixed forest with low ice content and a higher sand content or in locations where ice wedges are present and are likely melting. The mixed forest ecotypes have the warmest mean annual temperatures and are

thus at risk of thaw due to warming temperatures while the areas with ice wedges can exhibit dramatic subsidence as the ice melts.

Our study further confirms recent studies showing ERT measurements provide a robust way to characterize frozen versus thawed zones in permafrost terrains (Lewkowicz et al., 2011; Hubbard et al., 2013; Minsley et al; 2016; Douglas et al., 2008; 2016; Rey et al., 2020; Bjella, 2014; 2015; 2020). High resistivity areas identify permafrost

while low resistivity values correspond with thawed zones at the surface. Hotspots of low resistivity values correspond with deep active layer measurements, for example, at disturbed sites and across thermokarst features where thaw is identifiable from airborne imagery and LiDAR. Few studies have coupled ERT measurements with deep boreholes to corroborate the bottom or lateral extent of permafrost yet mapping these 0°C isotherm boundaries is critical for tracking and modelling lateral and top-down thaw. At the 358 m distance on the Farmer's

Loop 1 transect our borehole encountered a thawed zone at ~10m that corresponds exactly with the bottom of frozen soil measured by the large decrease in resistivity at that location (Fig. 4). Our ERT results show discontinuous permafrost is present at depths of up to at least 25 m across all transects but the lateral extent of small surface and subsurface thawed regions (as of 2014) are identifiable in ERT measurements at all sites.

The deep boreholes, in some cases representing the bottom of permafrost where bottom-up thaw is occurring

(McClymont et al., 2013) also provided access to wooden fragment samples amenable for [14]C dating. This allowed us to relate age and depth at three site locations. Based on the relationship between core depth and age date at the three locations where we have [14]C ages we calculate syngenetic permafrost deposition rates of 0.7 to 1 mm per year. This is close to the rates measured in the Permafrost Tunnel (Hamilton et al., 1988; T. Douglas, unpublished) and these deposition rates are important for mapping and modelling permafrost lateral and vertical extent across

remote locations.

## 5. Conclusions

The variety of measurements used in this study all confirm that near-surface permafrost around Fairbanks, Alaska has been undergoing dramatic warming and widespread thermokarst development since our measurements started





in 2013. The majority of the warming and thaw degradation are occurring in mixed forest ecotypes with low ice
content sandy silt soils, however, remote sensing evidence shows thermokarst features have been initiated in all
of the ecotypes represented. Though tussock tundra and spruce forest ecotypes are associated with the lowest
mean annual near surface permafrost temperatures at 1.2 m depth our results show warming below these ecotypes
as well. Thermistors at 1.2 m depth at the tussock and spruce forest sites reach the coldest winter temperatures of
all our sites so winter processes may be controlling the potential future thermal state of permafrost below these
ecotypes. Based on CALM site measurements, mapping, and geospatial analyses we conclude the rapid and
extensive thaw we identified at our field sites are common across the 500,000 km$^2$ area of central Alaska. Since
the yedoma type permafrost at our field sites and across the larger region is ice rich and has a high carbon content
there is high risk of thaw degradation and impacts to the carbon cycle. Based on our calculations the total central
Alaska yedoma permafrost organic carbon (OC) pool that has thawed since 2013 as 0.44 Gt OC. For perspective,
this is slightly more than the yearly $CO^2$ emissions of Australia (Friedlingstein et al., 2019). Results from this
study can support large scale modelling efforts on how current and projected future land cover will armour
permafrost against thaw and disturbance but also how and where ecotype changes can increase the risk of
permafrost thaw and thermokarst development. Our study sites are well suited to support these types of analyses
because the area contains warm permafrost, the climate has been warming since the 1970s, and our transects
represents most of the land cover present in the boreal and taiga of the Arctic and subarctic. The relationships we
found between ecotype, permafrost composition, and seasonal thaw dynamics can be used to apply biophysical
characteristics and standoff measurements like repeat aerial imagery, hyperspectral measurements, and LiDAR,
to ascertain the presence or absence of permafrost in similar terrains. This will help apply three-dimensional
thermal models to top down, lateral, and bottom up that of discontinuous permafrost bodies so future climate
projections can better be applied toward identifying the likely response of permafrost to warming.

*Data availability*

All project data are available through reasonable request.

*Author contribution*

*TAD, CAH, and JEA designed and initiated the study and performed field work; RAB, KLB, EJD, ABG, SDN,*
*SPS, and AMW assisted in the field and analysed data; PEN oversaw geospatial and mapping analyses; all*
*authors contributed to development and writing of this manuscript.*

*Competing Interests*





The authors declare no competing interests.

*Acknowledgements*

This research was funded by the U.S. Army Corps of Engineers Engineer Research and Development Center Basic Research Program under PE 0601102/AB2 (Protection, Maneuver, Geospatial, Natural Sciences) and Center Directed Research Program and the Department of Defense's Strategic Environmental Research and Development Program (Projects RC-2110 and RC18-1170). We thank Amanda Barker, Sam Beal, Marc Beede, Maria Berkeland, Dana Brown, Seth Campbell, Jarrod Edwards, Tiffany Gatesman, Malcom Major, Margaret Rudolph,

Torre Jorgenson, Merritt Turetsky, Simone Whitecloud, and Caiyun Zhang for help with field measurements. We appreciate comments about yedoma studies from Jens Strauss.



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



**Table 1. A summary of the thaw depth measurements by ecotype and results from a means comparison using a student's t-test. Among a given ecotype different and year the letters identify statistically significantly different means. Mean values for a given ecotype and year with similar letters have similar means.**

|  | Year | N | Mean | Standard deviation | Means comparison | % increase 2013-2020 |
|---|---|---|---|---|---|---|
| Tussock | 2013 | 126 | 45.0 | 10.9 | F |  |
|  | 2014 | 153 | 67.7 | 12.2 | D | 50 |
|  | 2015 | 153 | 63.3 | 12.6 | E | 41 |
|  | 2016 | 153 | 75.3 | 12.1 | A | 67 |
|  | 2017 | 153 | 72.2 | 13.8 | B, C | 60 |
|  | 2018 | 153 | 69.5 | 13.3 | C, D | 54 |
|  | 2019 | 153 | 72.8 | 13.9 | A, B | 62 |
|  | 2020 | 153 | 73.5 | 13.4 | A, B | 63 |
| Wetland | 2013 | 41 | 71.5 | 24.4 | D |  |
|  | 2014 | 48 | 91.4 | 23.5 | B, C | 28 |
|  | 2015 | 48 | 82.7 | 18.6 | C, D | 16 |
|  | 2016 | 48 | 104.9 | 30.0 | A, B | 47 |
|  | 2017 | 46 | 96.9 | 30.4 | B, C | 36 |
|  | 2018 | 47 | 103.1 | 39.2 | A, B | 44 |
|  | 2019 | 47 | 113.3 | 49.4 | A | 59 |
|  | 2020 | 47 | 113.6 | 51.5 | A | 59 |
| Disturbed | 2013 | 35 | 75.5 | 23.8 | C |  |
|  | 2014 | 67 | 85.8 | 20.4 | C | 14 |
|  | 2015 | 58 | 84.2 | 21.1 | C | 11 |
|  | 2016 | 60 | 99.9 | 28.8 | B | 32 |
|  | 2017 | 51 | 101.9 | 33.0 | B | 35 |
|  | 2018 | 56 | 104.3 | 35.0 | B | 38 |
|  | 2019 | 55 | 118.4 | 42.4 | A | 57 |
|  | 2020 | 56 | 117.1 | 44.3 | A | 55 |



| | Year | N | Mean | Standard deviation | Means comparison | % increase 2013-2020 |
|---|---|---|---|---|---|---|
| Mixed forest | 2013 | 57 | 64.4 | 17.9 | D | |
| | 2014 | 75 | 81.4 | 19.0 | B, C | 27 |
| | 2015 | 75 | 75.1 | 17.2 | C | 17 |
| | 2016 | 75 | 85.1 | 27.1 | B | 32 |
| | 2017 | 75 | 79.5 | 24.0 | B, C | 24 |
| | 2018 | 74 | 84.9 | 25.7 | B | 32 |
| | 2019 | 74 | 93.4 | 29.0 | A | 45 |
| | 2020 | 74 | 97.4 | 30.2 | A | 51 |
| Moss spruce | 2013 | 86 | 54.6 | 13.3 | E | |
| | 2014 | 111 | 59.7 | 10.7 | C, D | 9 |
| | 2015 | 120 | 56.5 | 12.6 | D, E | 3 |
| | 2016 | 118 | 64.0 | 11.6 | B | 17 |
| | 2017 | 124 | 62.5 | 12.2 | B, C | 14 |
| | 2018 | 115 | 64.9 | 14.8 | B | 19 |
| | 2019 | 119 | 70.1 | 16.0 | A | 28 |
| | 2020 | 115 | 72.5 | 18.7 | A | 33 |






**Table 2. A summary of thermistor measurements from 1.2m depth at the study site transects. Mean annual temperature (MAT) values for each of six individual years are presented as well as the six year global mean annual temperature for each site.**

| Creamer's Field, disturbed | | MAT °C | Creamer's Field, 1m-polygon center | | MAT °C | Creamer's Field, 1m- ice wedge in mixed forest | | MAT °C |
|---|---|---|---|---|---|---|---|---|
| 10/01/13 | 09/30/14 | 2.78 | 10/01/13 | 09/30/14 | -0.36 | 10/01/13 | 09/30/14 | -0.37 |
| 10/01/14 | 09/30/15 | 4.57 | 10/01/14 | 09/30/15 | -0.29 | 10/01/14 | 09/30/15 | -0.33 |
| 10/01/15 | 09/30/16 | 3.85 | 10/01/15 | 09/30/16 | -0.26 | 10/01/15 | 09/30/16 | -0.23 |
| 10/01/16 | 09/30/17 | 4.91 | 10/01/16 | 09/30/17 | -0.23 | 10/01/16 | 09/30/17 | -0.20 |
| 10/01/17 | 09/30/18 | 5.15 | 10/01/17 | 09/30/18 | -0.16 | 10/01/17 | 09/30/18 | -0.10 |
| 10/01/18 | 09/30/19 | 4.61 | 10/01/18 | 09/30/19 | -0.13 | 10/01/18 | 09/30/19 | -0.08 |
| 10/01/19 | 09/30/20 | N/A | 10/01/19 | 09/30/20 | -0.11 | 10/01/19 | 09/30/20 | -0.06 |
| | 6 year mean | 4.31 | | 7 year mean | -0.22 | | 7 year mean | -0.20 |
| Creamer's Field- mixed forest | | MAT °C | Creamer's Field- tussocks | | MAT °C | Farmer's Loop 1- mixed forest | | MAT °C |
| 10/01/13 | 09/30/14 | -0.72 | 10/01/13 | 09/30/14 | -2.85 | 10/01/13 | 09/30/14 | -0.21 |
| 10/01/14 | 09/30/15 | -0.20 | 10/01/14 | 09/30/15 | -3.03 | 10/01/14 | 09/30/15 | -0.08 |
| 10/01/15 | 09/30/16 | -0.23 | 10/01/15 | 09/30/16 | -1.63 | 10/01/15 | 09/30/16 | -0.07 |
| 10/01/16 | 09/30/17 | -0.15 | 10/01/16 | 09/30/17 | -0.34 | 10/01/16 | 09/30/17 | -0.04 |
| 10/01/17 | 09/30/18 | -0.13 | 10/01/17 | 09/30/18 | -0.51 | 10/01/17 | 09/30/18 | -0.02 |
| 10/01/18 | 09/30/19 | -0.12 | 10/01/18 | 09/30/19 | -1.31 | 10/01/18 | 09/30/19 | 0.00 |
| 10/01/19 | 09/30/20 | -0.11 | 10/01/19 | 09/30/20 | -1.15 | 10/01/19 | 09/30/20 | 0.06 |
| | 7 year mean | -0.24 | | 7 year mean | -1.55 | | 7 year mean | -0.05 |
| Farmer's Loop 2, 240m- tussocks | | MAT °C | Farmer's Loop 2, 245m- tussocks | | MAT °C | Permafrost Tunnel- spruce forest | | MAT °C |
| 10/01/13 | 09/30/14 | -2.29 | 10/01/13 | 09/30/14 | -3.70 | 10/01/13 | 09/30/14 | -0.74 |
| 10/01/14 | 09/30/15 | -2.62 | 10/01/14 | 09/30/15 | -3.20 | 10/01/14 | 09/30/15 | -1.17 |
| 10/01/15 | 09/30/16 | -0.68 | 10/01/15 | 09/30/16 | -2.16 | 10/01/15 | 09/30/16 | -0.40 |
| 10/01/16 | 09/30/17 | -1.45 | 10/01/16 | 09/30/17 | -2.98 | 10/01/16 | 09/30/17 | -0.98 |
| 10/01/17 | 09/30/18 | -0.63 | 10/01/17 | 09/30/18 | -1.80 | 10/01/17 | 09/30/18 | -0.28 |
| 10/01/18 | 09/30/19 | -0.55 | 10/01/18 | 09/30/19 | -2.21 | 10/01/18 | 09/30/19 | -0.27 |
| 10/01/19 | 09/30/20 | -1.51 | 10/01/19 | 09/30/20 | -2.16 | 10/01/19 | 09/30/20 | -0.25 |
| | 7 year mean | -1.39 | | 6 year mean | -2.60 | | 7 year mean | -0.58 |



**Figure 1. A Worldview 2 (© Digital Globe) satellite image of the area around Fairbanks, Alaska identifying the field site sites (colored regions) and transects (white lines) in this study.**


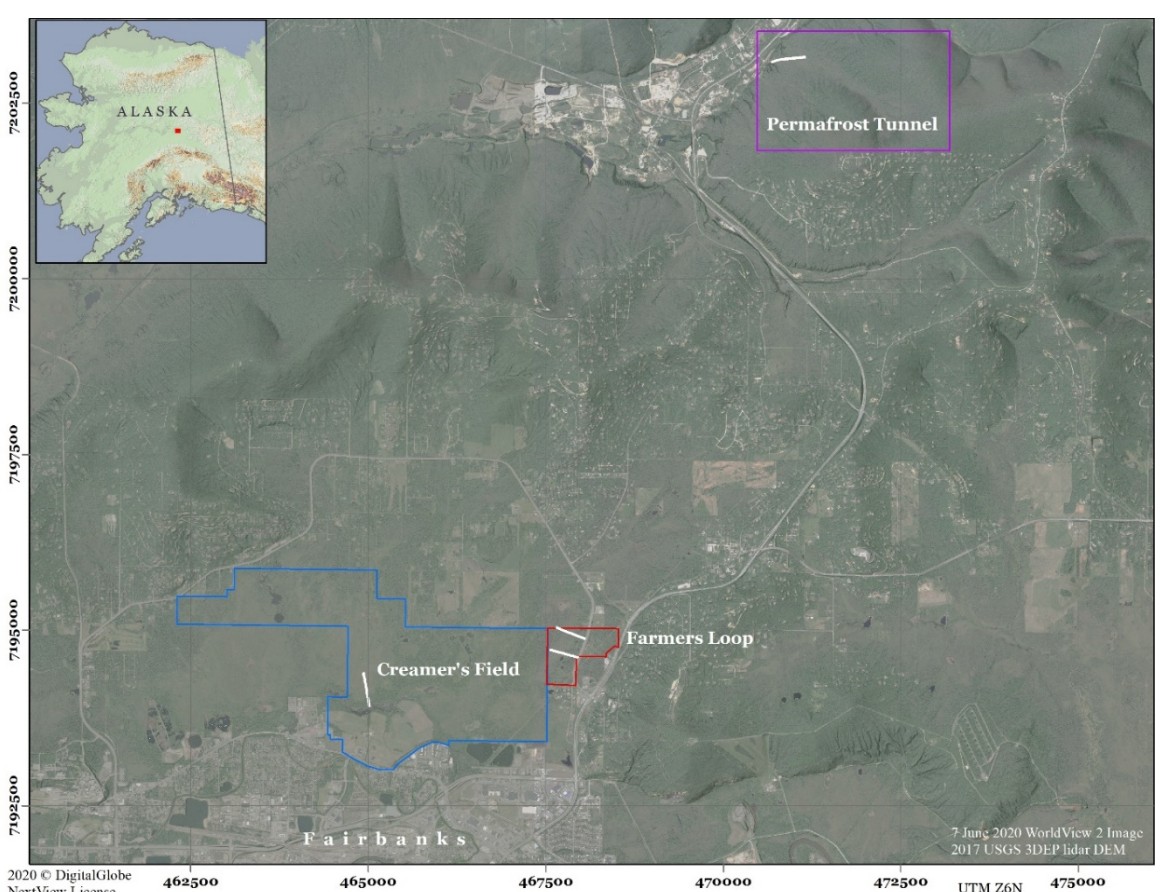



**Figure 2. The Creamer's Field transect from 0 to 246 m. A) Worldview 2 (© Digital Globe) true color image of the**
**transect (white line) with terrain features identified, B) LiDAR, C) repeat thaw depth measurements in 2014, D) repeat**
**active layer depth measurements from 2014-2019, E) a 246 m electrical resistivity tomography transect corrected for**
**ground surface elevation with boreholes identified as black boxes to true depth and numbers corresponding to the**
**distance (in meters) of the borehole location along the transect. "T" denotes a thermistor location.**





**Figure 3. The Creamer's Field transect from 252 to 498 m. A) Worldview 2 (© Digital Globe) true color image of the transect (white line) with terrain features identified, B) May 2020 LiDAR, C) repeat thaw depth measurements in 2014, D) repeat active layer depth measurements from 2014-2019, E) a 246 m electrical resistivity tomography transect corrected for ground surface elevation with boreholes identified as black boxes to true depth and numbers corresponding to the distance (in m) of the borehole location along the transect. "T" denotes a thermistor location.**








**Figure 4. The Farmer's Loop 1 transect. A) Worldview 2 (© Digital Globe) true color image of the transect (white line) with terrain features identified, B) May 2020 LiDAR, C) repeat thaw depth measurements in 2014, D) repeat active layer depth measurements from 2014-2019, E) a 410 m electrical resistivity tomography transect corrected for ground surface elevation with boreholes identified as black boxes to true depth and numbers corresponding to the distance (in m) of the borehole location along the transect. "T" denotes a thermistor location.**



**Figure 5. The Farmer's Loop 2 transect. A) Worldview 2 (© Digital Globe) true color image of the transect (white line) with terrain features identified, B) May 2020 LiDAR, C) repeat thaw depth measurements in 2014, D) repeat active layer depth measurements from 2014-2019, E) a 492 m electrical resistivity tomography transect corrected for ground surface elevation with boreholes identified as black boxes to true depth and numbers corresponding to the distance (in m) of the borehole location along the transect. "T" denotes a thermistor location.**






**Figure 6. The Permafrost Tunnel transect. A) Worldview 2 (© Digital Globe) true color image of the transect (white line) with terrain features identified, B) May 2020 LiDAR, C) repeat thaw depth measurements in 2014, D) repeat active layer depth measurements from 2014-2019, E) a 410 m electrical resistivity tomography transect corrected for ground surface elevation with boreholes identified as black boxes to true depth and numbers corresponding to the distance (in m) of the borehole location along the transect. "T" denotes a thermistor location.**





**Figure 7. Figure 7. Past (2010) elevations were subtracted from current (2020) elevations at the Creamer's Field site**
**(2020 minus 2010). The 500 m transect is denoted by the white line. Negative values identify regions of thaw degradation**
**and subsidence over the ten year period. Positive values show elevation gains due to deeper water and vegetation.**

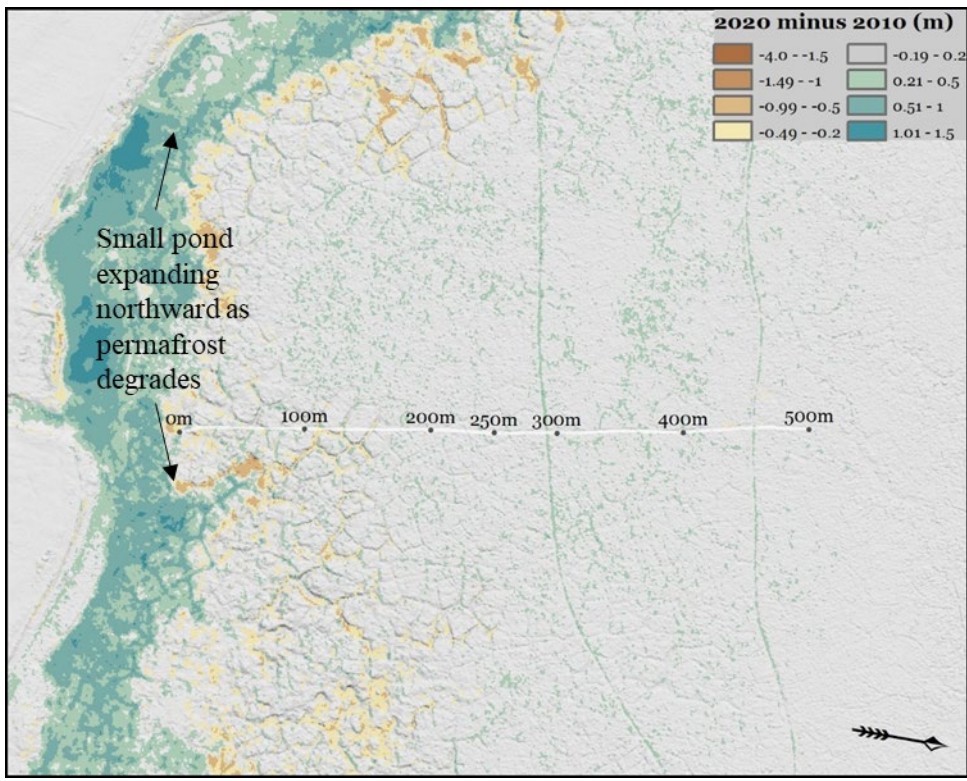



**Figure 8. Meteorologic conditions from April 1, 2013 through October 31, 2014. Top: air temperature from the Permafrost Tunnel and Farmer's Loop sites and historical daily mean values between 1929 and 2019 at the National Weather Service's Fairbanks International Airport (PAFA) site. Middle and bottom: precipitation at the PAFA site.**





**Figure 9. Soil temperature measurements at 1.2 m depth from October1, 2013 to October 1, 2019 for the three study sites. Mean annual ground temperature (MAGT) values at 1.2 m for the period of record are also provided.**



**Figure 10. Left- a © Google Earth map identifying yedoma type permafrost (yellow) in Alaska (Strauss et al., 2016) and locations of six central Alaska Circumpolar Active Layer Monitoring sites with records of at least 15 years. The focused field sites in this study are all near Farmer's Loop. The white bounding box represents the 500 km² area of central Alaska across which the study measurements are extrapolated. Right: active layer measurements from the six CALM sites from Interior Alaska and the Seward Peninsula. Data from CALM (2020).**

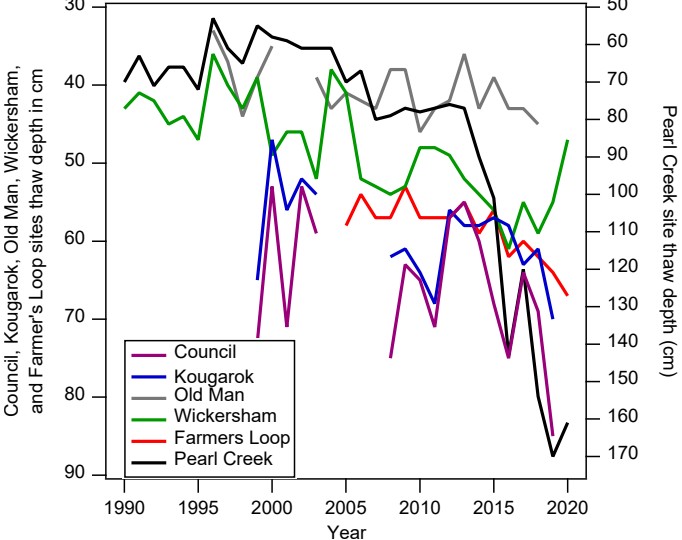

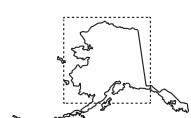