# Peer review of "Recent degradation of Interior Alaska permafrost mapped with ground surveys, geophysics, deep drilling, and repeat airborne LiDAR"

_The Cryosphere, 2021_

## Author Comment (AC1)

The paper by Douglas et al. represents a permafrost monitoring study that combines drilling, thermometry, geophysical and remote sensing approaches to detect permafrost dynamics over seven years in Central Alaska near Fairbanks in an area of discontinuous permafrost distribution. The combination of multiple belowground and aboveground observations allows the authors deducing significant interactions and trends between permafrost and surface features such as vegetation and disturbances over the observation period 2013-20. The permafrost response to current warming is clearly seen in the presented data and comprehensively interpreted by the authors. Thus, the vulnerability of discontinuously distributed permafrost to thaw under current warming is convincingly presented by Douglas et al. The impressive monitoring setup of drilling, ground temperature and thaw depth measurements combined with ERT, and satellite imagery and airborne LiDAR imagery at four different transects is, to my knowledge, exceptional in permafrost research.

The paper is clearly written and scientifically sound. To my opinion, the study by Douglas et al. represents a valuable contribution to The Cryosphere.

However, I suggest to restructure slightly the paper by adding a section 2 Study area (presently it is subsection 2.1) and a section 3 Material and methods. According changes throughout the manuscript are indicated below. Further, I noticed some redundancy in the paper and ask the authors to carefully shorten the text where appropriate. I have some further minor recommendations which are outlined below.

Thank you for this detailed review. We have reorganized the manuscript as suggested and address each comment in detail below.

p1 ln14: The abstract reads very detailed presenting much of the results of the study. I'd recommend some shortening while focusing on the main outcome of the study.

We have greatly reduced the abstract text as suggested.

The following sentences have been removed entirely:

"With a mean annual temperature of -2°C subtle differences in ecotype and permafrost ice and soil content control the near-surface permafrost thermal regime. Long-term measurements of the seasonally thawed "active layer" across central Alaska have identified an increase in permafrost thaw degradation that is expected to continue, and even accelerate, in coming decades."

We also shortened text toward the end of the abstract to:

"Our measurements, when combined with longer-term records from yedoma across the 500,000 $km^2$ area of central Alaska show widespread near-surface permafrost thaw since 2010. Projecting our thaw depth increases, by ecotype, across the yedoma domain we calculate 0.44 Gt of permafrost soil C have been thawed over the past 7 years, an amount equal to the yearly $CO_2$ emissions of Australia"

p1 ln24: Add space between '500' and 'm'.

This has been done:

"This study was conducted from 2013-2020 along four 400 to 500 m long transects near Fairbanks, Alaska."

p1 ln26: Define 'LiDAR' as your defined 'ERT' at first occurrence in the ms.
This has been done:
"Repeat end of season active layer depths, near-surface permafrost temperature measurements, electrical resistivity tomography (ERT), deep (>5 m) boreholes, and repeat airborne Light Distance and Ranging (LiDAR were used to measure top down thaw and map thermokarst development at the sites."

p1 ln30: Is '… active layer depth measurements must be made …' meant here?
Yes and the sentence has been changed to reflect this:
"At disturbed sites seasonal thaw increased up to 25% between mid-August and early October and suggests active layer depth measurements must be made as late in the fall season as possible because the projected increase in the summer season of just a few weeks could lead to significant additional thaw."

p2 ln38: Use subscripted number for '$CO_2$' here and elsewhere in the ms.
We apologize- it is unclear how this occurred. "$CO_2$" has been edited in both locations where it appeared. At the location in question:
"Projecting our thaw depth increases, by ecotype, across the yedoma domain we calculate 0.44 Gt of permafrost soil C have been thawed over the past 7 years, an amount equal to the yearly $CO_2$ emissions of Australia."

And in the Conclusions:
"For perspective, this is slightly more than the yearly $CO_2$ emissions of Australia (Friedlingstein et al., 2019)."

p3 ln46: Specify 'Mean annual air temperatures …'
We meant "mean annual air temperatures" and this has been changed to reflect that:
"Mean annual air temperatures in interior Alaska, currently roughly -2°C (Jorgenson et al., 2020), are projected to increase by 2°C by 2050 (Douglas et al., 2014) and 5 °C by 2100 (Lader et al., 2017)."

p3 ln53-54: Better use 'ice wedges' instead of 'massive ice bodies' since wedge ice is characteristic for Yedoma and you use this term anyway later on while massive ice is more general and might imply other origin.
Good point. Since our other studies have identified prevalent other massive ice features but ice wedges are indeed the most widespread we have edited the sentence to:
"Yedoma permafrost contains large organic carbon stocks that are extremely biolabile (Vonk et al., 2013; Strauss et al., 2017; Heslop et al., 2019) and highly vulnerable to thaw due to high ice content and the prevalence of massive ice bodies, particularly ice wedges (Strauss et al., 2013; 2017)."

p5 ln102-103: Do the area calculations refer to Strauss et al. (2016)? Add reference.
Yes this reference is applicable here and we have added it:
"Our sites represent 159,000 km$^2$ of high ice content yedoma permafrost with massive ice wedges that is present across the 500,000 km$^2$ expanse of central Alaska (Strauss et al., 2016)."

p5 ln 114: I'd suggest to omit section heading '2. Field measurements' and differentiate the second section as '2. Study area'. Consequently, the third section would start as '3. Material and methods' including the sub-sections '3.1 Satellite and LiDAR imagery', '3.2 Field survey measurements, coring and meteorology', '3.3 Electrical Resistivity Tomography'.

Thank you for this suggestion the Section numbering and labeling have been changed to:

**2. Study Area**

2.1 Study location and Site Descriptions

**3. Material and Methods**

3.1 Satellite and LiDAR imagery

3.2 Field survey measurements, coring, and meteorology

3.3 Electrical resistivity tomography

**4. Results**

4.1 Satellite and LiDAR imagery

4.2 Field survey measurements, coring, and meteorology

4.3 Electrical resistivity tomography

**5. Discussion**

**6. Conclusions**

p6 ln130: Add space between '400' and 'm'.

This has been edited, as suggested, to:

"The 400 m "Farmer's Loop 1" and 500 m "Farmer's Loop 2" transects were located at 64.877 °N, 147.674 °W and 64.874 °N, 147.677 °W, respectively."

p6 ln145-146: The statement is also given on p8 ln212 and could be omitted either here or there.

We have deleted the following sentence from lines 212-213:

"In total, these classes account for 74% of the boreal ecoregion's area in North America (Latifovic et al., 2017)."

p6 ln147: Insert section '3. Material and Methods'.

This has been done as suggested earlier.

p6 ln147: Rename sub-section heading to '3.1 …'.

This has been done as suggested earlier.

p6 ln156: Define 'DSM'.

This has been changed to:

"The resulting point cloud (point density = 4 points/m$^2$) was used to create a 1.21 m resolution digital surface model (DSM) with a vertical accuracy of 16 cm."

p7 ln162: Define 'DEM'.

This has been changed to:
"Hydro-flattening was used to remove errant point cloud elevation artefacts from resulting digital elevation models (DEMs) given water's low reflectance."

p7 ln169: Rename sub-section heading to '3.2 …'.
This has been done as suggested earlier.

p7 ln176: Delete space after 'August'.
This has been edited, as suggested, to:
"Measurements were repeated in mid-October from 2013 to 2020, however, in 2014 additional measurements were made in June, July, and August."

p8 ln191: Rename sub-section heading to '3.3 …'.
This has been done as suggested earlier.

p8 ln194: Delete dot after '4 m'.
This has been edited, as suggested, to:
"Our electrode spacings of 2.5 to 4 m achieved a maximum subsurface penetration depth of ~30 m."

p8 ln208: Rename section heading to '4. …'.
This has been done as suggested earlier.

p8 ln209: Rename sub-section heading to '4.1 …'.
This has been done as suggested earlier.

p8 ln212: The statement is also given on p6 ln145-146 and could be omitted either here or there.
We have deleted the following sentence from lines 212-213:
"In total, these classes account for 74% of the boreal ecoregion's area in North America (Latifovic et al., 2017)."

p9 ln216-217: I assume what you see in the LiDAR imagery and describe here are high-centre polygons that remain when the surrounding wedge ice melts. Consider omitting the term baydzherakh here and elsewhere in the manuscript. The correct term however would be thermokarst mound to skip the 'graveyard' nomenclature. Commonly acknowledged definitions on permafrost terminology can be found in van Everdingen, R.O. (1998) Multi-Language Glossary of Permafrost and Related Ground- Ice Terms; https://globalcryospherewatch.org/reference/glossary_docs/Glossary_of_Permafrost_and_Groun d-Ice_IPA_2005.pdf. Here, under no. 256 high-centre polygons and under no. 560 thermokarst mounds are defined.
This has been edited to:
"This area is characterized by high-centered polygons up to 2 m high that form when ice wedges melt."

And in (now) 4.2 Field survey measurements, coring, and meteorology:
"Notably, these include thawed zones in the high-centered polygons along the beginning of the Creamer's Field transect, thermokarst pits along the Farmer's
Loop and Permafrost Tunnel transects, a large (~50m) lateral expansion of the thermokarst

toward the end of the Permafrost Tunnel transect, and thawed regions below numerous disturbed areas at all sites."

And in (now) 4.3 Electrical resistivity tomography:
"In this area, pockets of low resistivity material in the upper 1-2 m denote the thawed areas around the high-centered polygons."

**And in two locations in (now) 5. Discussion:**
"Ice wedge polygons in the area have warmed steadily since 2013 and repeat LiDAR analysis shows development of high-centered polygons development has expanded due to melting ice wedges (Fig. 7)."

And:
"In this area near surface permafrost soils, comprised of lower ice content silts and sands, are warmer and high-centered polyons were already forming when we initiated our study"

p9 ln224: Add space between '310' and 'm'.
This has been edited, as suggested, to:
"Both transects start in mixed forest that extends for ~120 m. Transect 1 crosses a small wetland feature at 80 m before transitioning to tussock tundra until 310 m."

p9 ln225: Change to 'P. mariana'.
This has been edited, as suggested, to:
"After a graminoid-dominated trail the ecotype changes to *P. mariana* (black spruce) forest. Farmer's Loop transect 2 shifts from mixed forest (*B. neoalaskana*, *Salix* spp., and *P. glauca*) to a flow through fen wetland from 120 to 170 m.

p9 ln226: Change to 'B. neoalaskana' and 'P. glauca'.
This has been edited, as suggested (see above)

p9 ln229: Change to 'P. mariana'.
This has been edited, as suggested, to:
After a trail crossing at 200 m the ecotype shifts abruptly to tussock tundra until the 400 m mark where a trail crossing separates the tussock tundra from mature *P. mariana* (black spruce) forest."

p9 ln243: Delete second dot at the end of the sentence.
This has been edited, as suggested, to:
"Water levels in the pond and in some ice wedge polygon troughs show higher elevations in 2020 compared to 2010 (0.2 to 1.0 m) due to deeper and more persistent precipitation in the last three years."

p10 ln244: Rename sub-section heading to '4.2 …'.
This has been done as suggested earlier.

p10 ln245-246: Consider rephrasing to 'The mid-June and early August seasonal thaw depth measurements and those in October 2014 show …'.

We respectfully disagree with this suggested edit. It is important to identify the differences in terminology and meaning between thaw depth measurements made at any time during the summer thaw season and end of thaw season active layer measurements made in late fall. There is a lot of confusion about the difference between these two terms and their application. We find it is important to distinguish this difference because of the measurements we provide in Figures 2c, 3c, 4c, 5c, and 6c. We propose to identify that non-October measurements are that of seasonal thaw while mid-October measurements represent the active layer thickness/depth.

p10 ln257: Replace 'fifty' with '50'.
This has been edited, as suggested, to:
"The tussock, wetland, disturbed, and mixed forest ecotypes all exhibited increases in active layer depth of more than 50 percent while the increase in the spruce forest was 33 percent."

p10 ln272: Add 'at' to '358 m'.
This has been edited, as suggested, to:
 "Notably, a deep core on the Farmer's Loop 1 transect (at 358 m) collected thawed silt from 9.15-10.35 m."

p11 ln278: Add space between '50' and 'm'.
This has been edited, as suggested, to:
 "Notably, these include thawed zones in the high-centered polygons along the beginning of the Creamer's Field transect, thermokarst pits along the Farmer's Loop and Permafrost Tunnel transects, a large (~50 m) lateral expansion of the thermokarst toward the end of the Permafrost Tunnel transect, and thawed regions below numerous disturbed areas at all sites."

p11 ln287-290: Add a sub-section on radiocarbon dating to section 3 'Material and methods' and a table with the dating results (probably to the supplement) including sample ID, lab ID, sample depth, material, $\delta^{13}C$, sample mass, and radiocarbon age although some of this information is given as description in Table S1. Please, further consider common nomenclature and calibartion to give ages as years before present (yr BP) or calibrated years before present (cal yr BP).

We have added a new Section "4.3 Radiocarbon dating" and a new Section "4.4 Air and ground temperature measurements" as suggested.

We have edited the text to clarify "calibrated years before present (cal YBP) as follows:
"We obtained $^{14}C$ ages from wood fragments collected from three Geoprobe core samples through Geochron Laboratories (Chelmsford, Massachusetts, USA). An age of 10,360 +/- 360 calibrated years before present (cal. YBP) ($\delta^{13}C$: -27.7 ‰) was measured at a depth of 1.02 m in the tussock area at 306 m on Farmer's Loop transect 1 (Supplementary Table 1). At 358 m along the same transect and also in the tussock area at 0.67 m depth the $^{14}C$ age was 10,160 +/-160 cal. YBP ($\delta^{13}C$: -28.0 ‰). Along the Farmer's Loop 2 transect, in the spruce forest at 420 m and at a depth of 0.49 m depth a wood fragment yielded a $^{14}C$ age of 7,200 +/-190 cal. YBP ($\delta^{13}C$: -28.7 ‰)."

We have added/clarified all of the requested sample information to Table S1 but the analytical laboratory did not provide sample mass.

p12 ln316: Rename sub-section heading to '4.3 …'.

This has been done as suggested earlier but based on the comment above this is Section 4.5

p14 ln358: Rename section heading to '5. …'.
This has been done as suggested earlier.

p14 ln370: Add space between '1.2' and 'm'.
This has been edited, as suggested, to:
"The length of the summer growing season in the area has increased by 38 days (Wendler and Shulski, 2009) and our thermistor measurements (Figure 9) show peak soil temperatures at 1.2 m typically occur in late November."

p14 ln375: Consider rephrasing to '… have established that vegetation provides …'.
This has been edited, as suggested, to:
"Previous studies have established that vegetation provides a range of ecosystem protection properties for permafrost (Shur and Jorgenson, 2007; Loranty et al., 2018)."

p14 ln384: Add space between '2' and 'm'.
This has been edited, as suggested, to:
 "In many locations, active layer depths have increased since 2013 to greater than 2 m which is greater than typical winter freezeback."

p14 ln384: Delete space before 'Infrastructure …'.
This has been edited, as suggested.

p15 ln393: Add dot after 'al'.
This has been edited, as suggested, to:
"Our results support recent work at our study sites that show the disturbed, mixed forest, and wetland ecotypes exhibit the deepest active layers (Douglas et al., 2020)."

p16 ln430: Because ground subsidence and thermokarst are no synonyms, I'd suggest to remove the term thermokarst given in brackets after (ground subsidence).
This has been edited, as suggested, to:
"The most dramatic is that of the ground subsidence associated with permafrost thaw in the mixed forest region of the Creamer's Field transect (Fig. 9)."

p16 ln444: Consider rephrasing to '… measurements show that the thermokarst …'.
This has been edited, as suggested, to:
"Repeat yearly active layer depth measurements show that the thermokarst features have been extending vertically downward and horizontally since 2013."

p16 ln445: Add space between '100' and 'm'.
This has been edited, as suggested, to:
"Examples are at the Creamer's Field transect at ~100 m; Farmer's Loop transect 1 at 68 m and 360 m; Farmer's Loop transect 2 at 8 m, 88 m, 116 m, and 408 m; and the Permafrost Tunnel transect at 64 m, 108m, 140 m, and particularly at 328 m."

p17 ln447: Should read '108 m' instead of '10.8m'.
This has been done as suggested above.

p17 ln460: Add space between '10' and 'm'.

Thank you for identifying all these discrepancies in our distance measurements and labels. We apologize that there were so many inconsistencies. This has been edited, as suggested, to:

 "At the 358 m distance on the Farmer's Loop 1 transect our borehole encountered a thawed zone at ~10 m that corresponds exactly with the bottom of frozen soil measured by the large decrease in resistivity at that location (Fig. 4)".

p17 ln468: Delete 'T. Douglas, unpublished'.

This has been edited, as suggested, to:

 "This is close to the rates measured in the Permafrost Tunnel (Hamilton et al., 1988;) and these deposition rates are important for mapping and modelling permafrost lateral and vertical extent across remote locations."

p17 ln471: Rename section heading to '6. …'.

This has been done as suggested earlier.

p18 ln485: Use subscripted number for '$CO_2$' here and elsewhere in the ms.

This has been done as suggested earlier.

p18 ln496-497: Please, consider the data policy of the Copernicus journals: https://www.the-cryosphere.net/policies/data_policy.html

Thank you for this suggestion.

We uploaded the repeat seasonal thaw (2014) and repeat active layer (2014-2020), and yedoma area carbon content calculations to the following Zenodo doi:

10.5281/zenodo.4670463

The following text has been added to the statement on data availability:

"Repeat seasonal thaw depth measurements (2014), repeat active layer measurements (2014-2020), and yedoma area carbon content calculations are available through Zenodo using doi: 10.5281/zenodo.4670463. All project geophysical data are available through reasonable request."

p27 Table 1: Add 'cm' as depth unit to the 'mean' column.

This has been done as suggested.

p30 Figure 1: Consider rephrasing to 'Worldview 2 (© Digital Globe) satellite image of the area around Fairbanks, Alaska (red dot) identifying the field site sites (colored regions) and transects (white lines) in this study.

This has been done with a slight change ("red dot on inset map" instead of "red dot on map" to:

**"Figure 1. A Worldview 2 (© Digital Globe) satellite image of the area around Fairbanks, Alaska (red dot on inset map) identifying the field site sites (colored regions) and transects (white lines) in this study."**

p31 Figure 2, p32 Figure 3, p33 Figure 4 and p34 Figure 5, p35 Figure 6: Since the 'white line' refers to the transect shown in Figure 1, add 'white line in Fig. 1' in each of these captions. Sub-figures in the caption are identified by capitalized letters and 'non-capitalised' letters in the figure. Please adjust.

The white line has now been referred to in each caption and new versions of the Figures with capitalized sub-figure labels has been added.

p38 Figure 9: Consider fixed scale for the y-axes (temperatures) ranging from –12 °C to 12 °C to enhance the comparability of the nine T plots over time presented here.
We made many versions of these plots and when we present the with this broader y-axis scale (to be consistent) the subtle but small changes at locations where it is apparent latent heat is slowing down thaw are lost. Since we refer to this process in the text and feel it is an important aspect of the "warm" permafrost I the area we respectfully request to keep the y-axes scales as they currently are.

p39 Figure 10: In the caption, do you mean '500 km$^2$' or '500,000 km$^2$' as on p5 ln 103? Differentiate the combined Figure 10 into (a) and (b) instead of 'left and 'right'. Add scale and coordinates to the Alaska map (a). Here, location names whose data are presented in (b) are barely seen. In (b), consider one y-axis for thaw depth covering all locations presented here and ranging accordingly from 30 to 170 cm thaw depth.
In the caption we definitely mean 500,000 km$^2$. Thank you for identifying this discrepancy.
We have fixed the references to "left" and "right" and have presented the panels as a and b. The new caption reflecting this is:
**"Figure 10. A) a © Google Earth map identifying yedoma type permafrost (yellow) in Alaska (Strauss et al., 2016) and locations of six central Alaska Circumpolar Active Layer Monitoring sites with records of at least 15 years. The focused field sites in this study are all near Farmer's Loop. The white bounding box represents the 500,000 km$^2$ area of central Alaska across which the study measurements are extrapolated. B) active layer measurements from the six CALM sites from Interior Alaska and the Seward Peninsula. Data from CALM (2020)."**

When we plot the entire dataset of active layer measurements on a y-axis that ranges from 30 to 175 cm the trends at the Council, Kougarok, Old Man, Wickersham, and Farmer's Loop sites are lost due to the extreme range in values at the Pearl Creek site.

Table S1: Add coordinates of the drill locations. Unfortunately, cryostructures are not described, but probably beyond the focus of this study.
Latitude and Longitude location information for each core location has been added to Table S1

---

## Author Comment (AC2)

Dr. Thomas A. Douglas
U.S. Army Cold Regions Research and Engineering Laboratory
9th Avenue, Building 4070
Fort Wainwright, Alaska 99703
Phone: 907-361-9555; Fax: 907-361-5142
E-mail: Thomas.A.Douglas@usace.army.mil

Dear Copernicus Editor,                                                          April 15, 2021

We have edited the manuscript "Recent degradation of Interior Alaska permafrost mapped with ground surveys, geophysics, deep drilling, and repeat airborne LiDAR" by ten co-authors and myself.

The revised manuscript is substantially improved by the constructive comments of Reviewer Wetterich and an anonymous Reviewer and we thank them for their time and effort. Of particular note we shortened the abstract, reorganized the major sections and some sub-sections, and addressed numerous grammatical and typographical edits.

Our detailed addressal of the Reviewer 2's comments is provided below. Original Reviewer text is in black, our comments are in blue, and the revised/updated text is in red with quotation marks.

We look forward to hearing from you.

Sincerely,

Thomas A. Douglas

**RC2**: 'Comment on tc-2021-47', Anonymous Referee #2, 14 Apr 2021

The article by Douglas et al. incorporates long-term monitoring of ground temperatures and thaw depths with LiDAR-based analyses, radiocarbon dating, and permafrost coring. This is an extremely well-planned and thorough investigation of permafrost near Fairbanks, Alaska. The study is clearly presented and explained.

The study is appropriate for The Cryosphere, and will likely be of interests to scientists and policymakers across multiple disciplines.  In general, the conclusions are supported by the presented data, and the thawing and changes to the permafrost are clearly presented in many of the figures and tables. A very thorough, interesting and important study of Alaskan permafrost.

The only conclusion that seems like a reach is the calculation of potential carbon release for all of the yedoma-type permafrost in central Alaska. Given the many variables that can impact permafrost thaw and active layers (as well as variations in carbon-content even across yedoma-type permafrost), I question whether the authors can actually constrain that the total thaw of permafrost carbon in central Alaska to 0.44 Gt.

We thank the Reviewer for this constructive review and address each individual comment below.

Line 16 – long description is a bit confusing "ice rich high carbon content syngenetic yedoma permafrost"
We have broken this into two sentences for clarification:
"Of particular concern is thawing syngenetic "yedoma" permafrost which is ice rich and has a high carbon content . This type of permafrost is common in the region around Fairbanks, Alaska and in a region of Central Alaska that expands westward to the Seward Peninsula."

Line 24 – 500 m needs space between
This has been edited, as suggested, to"
"This study was conducted from 2013-2020 along four 400 to 500 m long transects near Fairbanks, Alaska."

Line 30 – I think "made" is the wrong word here. Confusing sentence.
The text has been changed to the following based on another Reviewer's suggestion and we feel it addresses this comment as well:
"At disturbed sites seasonal thaw increased up to 25% between mid-August and early October and suggests active layer depth measurements must be made as late in the fall season as possible because the projected increase in the summer season of just a few weeks could lead to significant additional thaw."

Line 38 – $CO_2$ subscript. Spell out carbon. "7-year"
The text has been changed to the following based on another Reviewer's suggestion and we feel it addresses this comment as well:
"Projecting our thaw depth increases, by ecotype, across the yedoma domain we calculate 0.44 Gt of permafrost soil C have been thawed over the past 7 years, an amount equal to the yearly $CO_2$ emissions of Australia."

Line 40 – comma after "cover"

This has been edited, as suggested, to:
 "Since the yedoma permafrost and the variety of ecotypes at our sites represent much of the Arctic and subarctic land cover, this study shows remote sensing measurements, top-down and bottom-up thermal modelling, and ground based surveys can be used predictively to identify areas of highest risk for permafrost thaw from projected future climate warming."

Line 129 – maybe write (n=3) and (n=1)
This has been edited, as suggested, to:
"Our field investigations were organized along four transects crossing a variety of lowland (n=3) and upland (n=1) permafrost landscapes (Fig. 1)."

Line 260 – where are the cores located on Figure 2–6. Can you include labeled sample circles?
We have added circles for core locations in Figures 2-6 and changed the caption accordingly.

Line 287 – are these radiocarbon years or calibrated years? Also, need to include the radiocarbon methods earlier in the article. Maybe include a table of the results.
This is similar to comments of another Reviewer which we addressed as follows:

We have added a new Section "4.3 Radiocarbon dating" and a new Section "4.4 Air and ground temperature measurements" as suggested.

We have edited the text to clarify "calibrated years before present (cal YBP) as follows:
"We obtained $^{14}$C ages from wood fragments collected from three Geoprobe core samples through Geochron Laboratories (Chelmsford, Massachusetts, USA). An age of 10,360 +/- 360 calibrated years before present (cal. YBP) ($\delta^{13}$C: -27.7 ‰) was measured at a depth of 1.02 m in the tussock area at 306 m on Farmer's Loop transect 1 (Supplementary Table 1). At 358 m along the same transect and also in the tussock area at 0.67 m depth the $^{14}$C age was 10,160 +/-160 cal. YBP ($\delta^{13}$C: -28.0 ‰). Along the Farmer's Loop 2 transect, in the spruce forest at 420 m and at a depth of 0.49 m depth a wood fragment yielded a $^{14}$C age of 7,200 +/-190 cal. YBP ($\delta^{13}$C: -28.7 ‰)."

We have added/clarified all of the requested sample information to Table S1 but the analytical laboratory did not provide sample mass.

Line 367 – "deep end?" What do you mean here?
We mean the deepest end of season active layer measurements and have changed the text to:
 "By mid-July the disturbed and mixed forest ecotypes exhibit the most seasonal thaw and these ecotypes have the deepest end of season active layer depth measurements."

Line 370 – What is the importance of 1.2 m? Why is this the depth that many of the measurements are from?
This depth represented "stable" permafrost in areas where we established temperature measurements. In most places where thermistors have been installed the areas are still frozen year round. Typical thaw depths across our field sites in areas of stable permafrost are less than a meter so 1.2 m is a depth for long term temperature measurements.

We have added/edited the following in section "**3.2 Field survey measurements, coring, and meteorology**" (which was renumbered based on suggestions of another Reviewer:

"Onset HOBO U23 Pro v2 two channel external temperature loggers were installed at depths of 1.2 m at nine locations across our field sites at locations where this represents permafrost."

Line 375 – add "that" between established and vegetation, if you want.
Another Reviewer made this same suggestion and it has been changed to:
"Previous studies have established that vegetation provides"

Line 378–9 – this sentence reiterates ideas that have already been stated. You could delete.
We respectfully request to keep this sentence in here because it reiterates a fundamental aspect of our work that we hope can inform remote sensing applications- ecotype:thaw depth and ecotype:permafrost relationships can help support broader scale assessments.

Line 384 – Just during this period of time (after 2013)? Were active layers deepening before this?
This is in specific relation to our field site locations and measurements so this has been clarified to:
"In many locations at our field sites, active layer depths have increased since 2013 to greater than 2 m which is greater than typical winter freezeback."

Line 390 – "would increase" instead of increases.
This has been changed, as suggested, to:
"As such, if vegetation were to change from tussocks or spruce to a mixed forest or disturbed (i.e. no moss or forest vegetation) land cover the potential risk of top-down permafrost thaw would increase considerably."

Line 403 – what is "cookie cut"? Please explain.
This is a term used by some in the GIS community that is likely more commonly understood by non-GIS people in lieu of the word "clip." We have clarified it to:
"Using the regions mapped as yedoma as a "cookie cut" clipping from ecotype maps that cover all of central Alaska (Jorgenson and Meidlinger, 2015; Raynolds et al., 2019) we calculate the five ecotypes in our study represent 90% of the total land cover on top of yedoma permafrost."

Line 407–411 – Given the small region of the study and the variability in thaw dynamics across the region, I question whether the authors can constrain the organic carbon pool and potential loss from thawing across all of central Alaska. I think a better way to discuss this would be to calculate the total thawed permafrost and the organic carbon release from the study sites. They could then discuss how much more yedoma-type permafrost exists in Alaska – eluding to the potential magnitude of permafrost thaw, but not directly calculating it.
We developed this aspect of our research in an attempt to more broadly apply the results from our specific site scale measurements to a larger area of interest. This was partly to address comments made on an earlier version of the manuscript by Editor Morse. He suggested we provide a larger context than just our study sites. We feel the way this is presented limits over-simplification and adds impact to the paper. Few studies have identified links between top down thaw and potential carbon stocks. However, many studies have been synthesized to provide yedoma permafrost carbon stocks. We want to clarify that we have not measured carbon emissions from our sites. Also much of the Central Alaska region is extremely remote and there are few study sites or measurements. However the CALM sites across the region yield somewhat consistent results as our findings and this suggests the broader application we undertake.

We fully realize that we have to be careful in the broader application of our results and we have made an attempt to make sure we use generalized data and application. We are focused on a general calculation and assessment of the potential carbon that has been thawed. There is no mention of carbon cycle or gas production processes. We just identify the potential stocks thawed and relate them to a value (Australian emissions) that is easy to comprehend than the number itself. We are not weighing in on the fate of this carbon in soils or the atmosphere as part of the potential permafrost carbon feedback. As such, we feel that the way we have framed and presented this limits any direct assessment of greenhouse gas emissions. We note the other Reviewer did not make any comments to change this aspect of the manuscript and we respectfully request to keep this aspect of our Conclusions as it is.

Line 439 – does this decrease in elevation refer to all troughs? Is this the average? Was this calculated with LiDAR? Based on Fig. 7, it looks like trough subsidence only occurred near the lake/river – is there spatial variability?
This is a good point. We have clarified the text as follows:
"Some of the low lying troughs between polygons, particularly those along the thaw front next to the ponded area to the west, have dropped by 1-1.5 m in the decade from 2010 to 2020."

Line 442 – delete comma between May 2020
This has been edited, as suggested, to:
"We attribute this to this area being extremely low lying and more standing water being present in the troughs from snowmelt in May 2020 compared to 2010"

Line 447 – 108 or 10.8 or 10 8?
This was identified by another Reviewer. We have changed the text to:
"Examples are at the Creamer's Field transect at ~100 m; Farmer's Loop transect 1 at 68 m and 360 m; Farmer's Loop transect 2 at 8 m, 88 m, 116 m, and 408 m; and the Permafrost Tunnel transect at 64 m, 108 m, 140 m, and particularly at 328 m."

Line 487–480 – This sentence is confusing.
The word "represent" should not have bene plural. This has been changed to:
"Our study sites are well suited to support these types of analyses because the area contains warm permafrost, the climate has been warming since the 1970s, and our transects represent most of the land cover present in the boreal and taiga of the Arctic and subarctic."

Figure 2–6 – the caption references "white line" but there is no white line in the image.
The white line reference is for the white lines in Figure 1. Another Reviewer identified this same issue as well as other suggested Figure caption edits. We have edited the Figure captions for Figures 2-6 to (as an example):
**"Figure 2. The Creamer's Field transect from 0 to 246 m (white line in Fig. 1). Image a) is a Worldview 2 (© Digital Globe) true color image of the transect (white line) with terrain features and core locations (circles) identified, b) LiDAR, c) repeat thaw depth measurements in 2014, d) repeat active layer depth measurements from 2014-2019, and e) a 246 m electrical resistivity tomography transect corrected for ground surface elevation with boreholes identified as black boxes to true depth and numbers corresponding to the distance (in meters) of the borehole location along the transect. Stars with a "T" denote a thermistor location."**

Figure 2–6 – please show core sites
Core sites have been added to panel a and the caption has been edited to reflect this.

Figure 9 – in caption put space between October 1
Thank you for catching this. It has been changed to:
**"Figure 9. Soil temperature measurements at 1.2 m depth from October 1, 2013 to October 1, 2019 for the three study sites. Mean annual ground temperature (MAGT) values at 1.2 m for the period of record are also provided."**

Figure 10 – try to make the locations and labels easer to see on the map of Alaska.
The locations and labels have been edited to make them clearer.

---

## Editor Decision (ED1)

[revised manuscript text omitted]

64.8753N, 147.6854W

| Distance: 431 m | Start (cm) | End (cm) | Gravimetric moisture content (g/g) % | Frozen bulk density (kg/m³) | Description |
|---|---|---|---|---|---|
| 5/3/2017 | 0 | 6 | 69.4 | 815 | Organics with peat |
| SIPRE | 6 | 21 | 88.6 | 1145 | Organics with peat |
| | 21 | 34 | 78.4 | 1143 | Organics with peat |
| | 34 | 45 | 105.6 | 1436 | Organics with peat |
| | 45 | 59 | 65.0 | 1180 | Organics with peat |
| | 59 | 72 | 93.4 | 1188 | Silt with visible ice |
| | 72 | 84 | 91.7 | 1338 | Silt with visible ice |
| | 84 | 106 | 55.8 | 1097 | Silt with no visible ice |
| | 106 | 118 | 70.0 | 1591 | Silt with no visible ice |
| | 118 | 132 | 59.4 | 1394 | Silt with no visible ice |
| | 132 | 147 | 50.6 | 1494 | Silt with no visible ice |
| | 147 | 161 | 55.6 | 1565 | Silt with no visible ice |
| | 161 | 175 | 58.9 | 1617 | Silt with no visible ice |
| | 175 | 188 | 63.6 | 1500 | Silt with no visible ice |
| | 188 | 200 | 81.9 | 1482 | Silt with no visible ice |
| | 200 | 220 | 40.6 | 1078 | Silt with no visible ice |

**Permafrost Tunnel Transect**

64.9507N, 147.6196W

| Distance: 52 m | Depth start (cm) | Depth finish (cm) | Description |
|---|---|---|---|
| 6/5/2018 | 0 | 17 | Thawed peat |
| SIPRE | 17 | 31 | Frozen silt with visible ice |
| | 31 | 49 | Muddy thawed silt |
| | 49 | 74 | Silt with visible ice |
| | 74 | 100 | Silt with visible ice |

64.9510N, 147.6141W

| Distance: 305 m | Depth start (cm) | Depth finish (cm) | Description |
|---|---|---|---|
| 3/25/2018 | 0 | 15 | Peat |
| SIPRE | 15 | 45 | Frozen silt with visible ice |
| | 45 | 70 | Muddy thawed silt |
| | 70 | 249 | Silt with visible ice |

64.9511N, 147.6122W

| Distance: 402 m | Depth start (cm) | Depth finish (cm) | Gravimetric moisture content (g/g) % | Description |
|---|---|---|---|---|
| 3/5/2014 | 75 | 90 | 91.4 | Silt with no visible ice |
| Geoprobe | 150 | 165 | 148.8 | Silt with no visible ice |
| | 216 | 232.5 | 119.7 | Silt with visible ice |
| | 240 | 255 | 145.9 | Silt with visible ice |
| | 345 | 360 | 96.5 | Silt with visible ice |
| | 420 | 435 | 23.6 | Silty sand |
| | 495 | 510 | 77.7 | Silt with visible ice |
| | 585 | 600 | 57.5 | Silt with visible ice |
| | 645 | 660 | 83.3 | Silt with visible ice |
| | 720 | 735 | 57.2 | Silt with visible ice |
| | 930 | 945 | 53.4 | Silt with visible ice |
| | 1020 | 1035 | 85.9 | Silt with visible ice |
| | 1125 | 1140 | 94.1 | Silt with visible ice |
| | 1230 | 1245 | 66.8 | Silt with visible ice |
| | 1380 | 1395 | 56.3 | Silt with visible ice |
| | 1485 | 1500 | 37.3 | Silt with visible ice |
| | 1515 | 1530 | 25.3 | Silt with visible ice |
| | 1545 | 1560 | 40.7 | Silt with visible ice |

---

## Author Response (AR2)

Editor Morse-

Thank you for the additional review and edits to the manuscript. You caught a lot of small grammar and spelling issues but also provide some suggestions to make the language simpler, shorter, and clearer. All of the suggested edits have been made, however, a few are discussed in detail below as I wanted to provide some additional perspective.

There is some confusion from me about what words to hyphenate or not (ice-rich, top-down, active-layer) as some people do hyphenate them and some do not. I have changed to ice-rich (ice-poor, etc.) and top-down but not active-layer. I see more papers use "active layer" than "active-layer". This includes papers in The Cryosphere but I will defer to you or the journal if it is preferred it be hyphenated. I have never hyphenated it in a paper or report or book chapter.

Comments specific to your edits/comments:

Line 22: we are reporting repeat end of season active layer depths. I want to make it clear that unlike a lot of studies that are done in the summer these are truly active layer measurements. I would like to keep them presented as such. I changed the text to:

"Repeat end of summer thaw season active layer depths, near-surface permafrost temperature measurements, electrical resistivity tomography (ERT), deep (>5 m) boreholes, and repeat airborne Light Distance and Ranging (LiDAR) were used to measure top-down permafrost thaw and map thermokarst development at the sites."

We do provide a set of "thaw depths" for the 2014 dataset so we could say "repeat seasonal thaw and end of season active layer depths" but I respectfully request we keep "active layer" in there somewhere.

Line #s in the below are keyed to the line numbers in your edited .pdf.

Line 28: I rarely see "active-layer" with the hyphenation. Please clarify if you want us to present it that way here and throughout the manuscript.

Lines 128-138: I am not clear based on the .pdf formatting what is preferred for the degrees, latitudes, longitudes, etc. The .pdf editing tool is hard to decipher with a lot of edits in a short text string. I tried my best.

Lines 154-165: DSMs are not appropriate here. The data were DEMs, representing bare earth, and those were the surfaces that were differenced. DSMs include bare earth + vegetation, like trees, and those points in canopies were filtered out. DEMs are more powerful for this analysis.

Please also see clarification for the 2010 versus 2020 scales added/edited as requested.

Line 223: I am not clear where these words are used again that they need to be cut/simplified. Do you mean here or in subsequent text? I think you mean here on so I edited the text to:

"After a graminoid-dominated trail the ecotype changes to black spruce forest. Farmer's Loop 2 transect shifts from mixed forest to a flow through fen wetland from 120 to 170 m. After a trail crossing at 200 m the ecotype shifts abruptly to tussock tundra until the 400 m mark where a trail crossing separates the tussock tundra from mature black spruce forest."

Line 233: "Numerous trail crossings identified as disturbed locations and a large thermokarst feature near Glenn Creek are also present."
It is suggested to change this to:
"Numerous trail crossings identified as disturbed locations and a large thermokarst feature near Glenn Creek is also present."

I am not the sure "is" is appropriate here.
How about:
"Numerous trail crossings are visible and are identified as disturbed locations and a large thermokarst feature near Glenn Creek is also present."

246-254: again it is more common to see "active layer" than "active-layer" Please advise if you really want this hyphenated.

263: if we are going to change ice rich" to "ice-rich" then should we change "ice poor" to "ice-poor"?

374: It is ok to remove this but it addresses the comment on the length of the summer season increasing.

409: it is the total increase not the average

429: There are taliks at many of these sites and though I agree we cannot be sure taliks are everywhere I request that we make that claim and acknowledge it. The seasonal freeze never goes more than 1.5 or maybe 2 m so in many locations it is clear these are not one-off warm summers or warm winter.

I have added this sentence at the end:
"Since thaw depths increased in 2019 and 2020 it is likely  residual thaw layers have increased in thickness and lateral extent. At locations where the thaw front has extended below 1.5 to 2 m it is likely that taliks (unfrozen zones between the bottom of the active layer and the top of the near-surface permafrost table) have formed."

Let me know is this is acceptable.

447-454: ok we can delete this. It was an attempt to provide specific locations of the features but if brevity is sought that can be removed.

---

## Editor Decision (ED2)

[revised manuscript text omitted]

---

## Author Response (AR3)

Editor Morse-                                                                                          May 31, 2021

The most recently uploaded version of the paper addresses all the edits and comments. We have provided below our response to the earlier version to ensure those comments of ours are seen.

Tom Douglas

Thank you again for another thorough reading of the paper.

I incorporated all your edits/suggestions, however, I wanted to comment about the use of the word "talik." In the previous version, you requested we remove all use of the word "talik" and instead say "residual thaw layers" or "unfrozen zones between the bottom of the active layer and the top of near-surface permafrost."

I request that we add back some application of the word "talik" in a few locations as our drilling confirms that these are taliks and I would like to use proper terminology instead of a long explanation. I have spoken with a few other researchers working at or near our sites (Torre Jorgenson and Vladimir Romanovsky) and all confirm that yes indeed there are taliks forming throughout the area.

The locations where I request we add back "talik" are:
In the abstract, line 25:
"Thermokarst features, residual thaw layers, and taliks have been identified at all sites."
My sense is that in some winters the active layer does not always refreeze completely at all sites. However, the areas with thaw confirmed from drilling or where that is greater than the ~1.2 to 1.5 m maximum depth of winter freeze taliks are no doubt forming.

At line 287:
"These thawed zones are located above the typical permafrost table for those locations and indicate the active layer did not completely freeze back and taliks are likely forming."

I also would like to add this at the end of the acknowledgements:
"We appreciate comments about yedoma studies from Jens Strauss and acknowledge two diligent reviewers and Editor Morse for their constructive comments on multiple versions of the manuscript."